# Diverse states and stimuli tune olfactory receptor expression levels to modulate food-seeking behavior

Ian G McLachlan[1], Talya S Kramer[1,2†], Malvika Dua[1†], Elizabeth M DiLoreto[3], Matthew A Gomes[1], Ugur Dag[1], Jagan Srinivasan[3], Steven W Flavell[1*]

[1]Picower Institute for Learning and Memory, Department of Brain and Cognitive Sciences, Massachusetts Institute of Technology, Cambridge, United States; [2]MIT Biology Graduate Program, Massachusetts Institute of Technology, Cambridge, United States; [3]Department of Biology and Biotechnology, Worcester Polytechnic Institute, Worcester, United States

**\*For correspondence:**
flavell@mit.edu

†These authors contributed equally to this work

**Competing interest:** The authors declare that no competing interests exist.

**Abstract** Animals must weigh competing needs and states to generate adaptive behavioral responses to the environment. Sensorimotor circuits are thus tasked with integrating diverse external and internal cues relevant to these needs to generate context-appropriate behaviors. However, the mechanisms that underlie this integration are largely unknown. Here, we show that a wide range of states and stimuli converge upon a single *Caenorhabditis elegans* olfactory neuron to modulate food-seeking behavior. Using an unbiased ribotagging approach, we find that the expression of olfactory receptor genes in the AWA olfactory neuron is influenced by a wide array of states and stimuli, including feeding state, physiological stress, and recent sensory cues. We identify odorants that activate these state-dependent olfactory receptors and show that altered expression of these receptors influences food-seeking and foraging. Further, we dissect the molecular and neural circuit pathways through which external sensory information and internal nutritional state are integrated by AWA. This reveals a modular organization in which sensory and state-related signals arising from different cell types in the body converge on AWA and independently control chemoreceptor expression. The synthesis of these signals by AWA allows animals to generate sensorimotor responses that reflect the animal's overall state. Our findings suggest a general model in which sensory- and state-dependent transcriptional changes at the sensory periphery modulate animals' sensorimotor responses to meet their ongoing needs and states.

## Editor's evaluation

This article provides a detailed explanation of how *C. elegans* adapts its behavior and chemosensory responses to shifts in food availability. The authors show that prolonged fasting broadly alters gene expression in food-sensing neurons, thereby altering foraging behavior and chemosensory responses to food. The fasting-induced genes include many chemoreceptors, one of which mediates responses to specific volatile components of food. Finally, they show that food controls expression of a fasting-induced chemoreceptor via multiple external (i.e., sensory) and internal (potentially metabolic) cues.

## Introduction

To thrive in a dynamic environment, animals must continuously integrate their perception of the outside world with their internal needs and experiences. Thus, virtually all animals exhibit long-lasting internal

states that reflect their physiology and experience and in turn influence sensorimotor processing. For example, an animal's nervous system might respond to the recent absence of food sensory cues and a metabolic energy deficit by generating a neural representation of hunger, which in turn alters a wide range of feeding-related behaviors. In real-world environments, an animal's nervous system is responsible for evaluating a number of needs simultaneously to prioritize behavioral outputs – such as avoiding predation while seeking food – to ensure that all needs are ultimately met. The biological underpinnings of these complex, integrated state-dependent behavioral changes remain poorly understood.

Decades of experimental work has identified neuronal populations that can induce internal states (*Flavell et al., 2022*). For example, NPY/AgRP neurons in the hypothalamus drive behavioral changes typical of hunger (*Aponte et al., 2011*; *Clark et al., 1984*; *Krashes et al., 2011*; *Luquet et al., 2005*), neurons in the lamina terminalis drive those typical of thirst (*Johnson and Gross, 1993*; *Oka et al., 2015*), and subpopulations of neurons in the ventromedial hypothalamus drive aggressive behaviors (*Kruk et al., 1983*; *Lin et al., 2011*). Likewise, P1 interneurons in *Drosophila* can trigger a state of social arousal (*Hindmarsh Sten et al., 2021*; *Hoopfer et al., 2015*), and serotonergic NSM neurons in *Caenorhabditis elegans* can trigger dwelling states during foraging (*Flavell et al., 2013*; *Ji et al., 2021*; *Rhoades et al., 2019*; *Sawin et al., 2000*). These devoted cell populations appear to respond to state-relevant inputs and elicit a suite of behavioral changes that comprise the state. However, animals can exhibit more than one state at a time, like hunger, stress, or aggression. Therefore, the sensorimotor pathways that implement specific motivated behaviors, such as approach or avoidance of a sensory cue, must integrate information about multiple states to adaptively control behavior. Previous work has revealed that neuromodulators and hormones can convey state information to sensory circuits to allow for state-specific sensorimotor processing (*Horio and Liberles, 2021*; *Inagaki et al., 2014*; *Jourjine et al., 2016*; *Ko et al., 2015*; *Root et al., 2011*; *Sayin et al., 2019*; *Takeishi et al., 2020*; *Yapici et al., 2016*), but how diverse state-related inputs are integrated by these circuits remains unclear.

The nematode *C. elegans*, whose nervous system consists of 302 defined neurons with known connectivity (*White et al., 1986*; *Witvliet et al., 2021*), exhibits a wide range of state-dependent behavioral changes (*Flavell et al., 2020*). Food deprivation leads to a suite of behavior changes, such as exaggerated dwelling and increased feeding rates upon encountering food (*Avery and Horvitz, 1990*; *Ben Arous et al., 2009*; *Sawin et al., 2000*; *Shtonda and Avery, 2006*); harmful stimuli can trigger states of generalized aversion or stress-induced sleep (*Chew et al., 2018*; *Hill et al., 2014*); and infection by a bacterial pathogen can trigger bacterial avoidance and changes in bacterial preference (*Kim and Flavell, 2020*; *Meisel et al., 2014*; *Zhang et al., 2005*). The *C. elegans* neuromodulatory systems (*Bentley et al., 2016*) allow these states to influence sensorimotor circuits: the effects of hunger are mediated by amines and insulin signaling (*Ghosh et al., 2016*; *Skora et al., 2018*; *Takeishi et al., 2020*); stressors induce the release of tyramine and neuropeptides that alter behavior (*De Rosa et al., 2019*; *Nath et al., 2016*; *Nelson et al., 2014*); and bacterial infection induces the release of *daf-7*/TGFβ, which promotes bacterial lawn leaving (*Meisel et al., 2014*). The well-defined behavioral states of *C. elegans*, together with its relatively simple sensorimotor circuits, make it an attractive system to decipher how animals generate sensorimotor behaviors that reflect an integration of their states.

Sensorimotor processing in *C. elegans* originates in primary sensory neurons that detect odorants, tastants, touch, temperature, and more (*Iliff and Xu, 2020*). Each of the 16 chemosensory neuron pairs expresses a multitude of chemoreceptors, which are predominantly G-protein-coupled receptors (GPCRs) (*Ferkey et al., 2021*). Detection of odorants or tastants evokes changes in sensory neuron activity that are transmitted to downstream interneurons and motor circuits (*Chalasani et al., 2007*; *Suzuki et al., 2008*). Several chemosensory neurons, such as AWA and AWC, primarily detect appetitive cues, while others, including ASH, detect aversive cues. However, chemosensory processing can be modulated by internal state and learning (*Flavell and Gordus, 2022*). For example, insulin signaling drives a hunger-dependent switch in thermotaxis behavior by modulating the AWC sensory neuron (*Takeishi et al., 2020*). Associative learning can drive changes in salt or temperature preference by altering presynaptic release from the ASE and AFD neurons that detect these respective stimuli (*Hawk et al., 2018*; *Ohno et al., 2017*). Neuromodulation of sensory interneurons also impacts sensory processing (*Chen et al., 2017*). In addition, internal states have been shown in some cases to

modulate gene expression in chemosensory neurons. Starvation alters the expression of the *str-234* chemoreceptor in ADL (*Gruner et al., 2014*) and the diacetyl receptor *odr-10* in AWA (*Ryan et al., 2014*; *Wexler et al., 2020*). Infection and starvation can modulate *daf-7*/TGFβ expression in ASJ (*Hilbert and Kim, 2017*). This work suggests that changes in chemosensory neuron gene expression are well-poised to underlie state-dependent changes in sensorimotor processing and likewise represent a plausible locus of state integration.

Here, we show that the olfactory neuron AWA integrates multiple streams of information to regulate chemoreceptor expression and dictate state-dependent food-seeking behavior. State-dependent ribotagging reveals that the expression of chemoreceptor genes is disproportionately elevated in AWA following food deprivation. We find that AWA chemoreceptor expression is controlled by both the sensory and metabolic components of food, as well as physiological stress. The state-dependent chemoreceptor *str-44* confers responsiveness to the putative food odors butyl acetate and propyl acetate and promotes starved-like foraging behaviors when expressed at high levels in AWA. Further, we delineate the neural and molecular pathways that underlie convergent signaling to AWA, identifying signaling pathways from other sensory neurons to AWA, a gut-to-brain metabolic pathway that signals to AWA, and a physiological stress pathway. These pathways act in a modular fashion and each contribute independently to the levels of AWA chemoreceptor expression. Our results reveal how diverse external and internal cues – nutritional state, stress, and sensory environment – converge at a single node in the *C. elegans* nervous system to allow for an adaptive sensorimotor response that reflects a complete integration of the animal's states.

## Results

### Diverse external and internal cues regulate chemoreceptor expression in the AWA olfactory neuron

As an unbiased approach to identify neural mechanisms that underlie state-dependent behavioral changes, we performed molecular profiling of *C. elegans* neurons in well-fed versus food-deprived animals. We selected 3 hr of food deprivation as our time point because it is sufficient to induce many feeding-related behavioral changes, including alterations in food approach, encounter, and exploitation (*Rhoades et al., 2019*), while remaining a relatively mild metabolic insult. To obtain a 'snapshot' of gene expression while animals were in a specific state, we used pan-neural ribotagging. This method permits cell-specific purification of actively translating mRNAs from animals that are flash frozen within minutes after removal from plates. We expressed an HA-tagged ribosomal subunit in all neurons, purified the tagged mRNA-ribosome complexes from fed and 3-hr fasted adult animals, and sequenced the isolated mRNAs and whole-animal input RNA (see 'Materials and methods'; *McLachlan and Flavell, 2019*). High-depth mRNA sequencing of ribotag samples allowed us to detect mRNAs that were being actively translated in as few as a single pair of neurons. We performed three independent biological replicates (*Figure 1—figure supplement 1A*) and confirmed the enrichment of pan-neural mRNAs in ribotag samples (*Figure 1—figure supplement 1B*). We examined the genes whose expression was most dramatically altered by fasting: 802 genes were increased by more than fourfold in the fasted condition compared to the fed condition, while 647 were decreased by more than fourfold (*Figure 1A*; *Supplementary file 1*). Strikingly, chemosensory GPCRs were significantly overrepresented among the upregulated genes (133/802 upregulated genes, p<0.001, Fisher's exact test, compared to ~8.5% overall prevalence in the genome; *Figure 1—figure supplement 1C*). These results suggest that food deprivation causes an upregulation of chemosensory GPCRs in *C. elegans* neurons.

As a first step to identify the neurons where the fasting-induced chemosensory GPCRs are expressed, we examined the site(s) of expression of these 133 genes in publicly available single-cell sequencing data (*Taylor et al., 2021*). We found that these chemoreceptors are distributed across all neurons in the amphid sense organ, suggesting that fasting induces broad changes in olfactory coding by sensory neurons (*Figure 1B*, gray bars). To determine whether any sensory neurons exhibit a disproportionate change in chemoreceptor expression profile after fasting, we normalized the counts of upregulated receptors to the overall number of chemosensory GPCRs expressed by each neuron. We found that the AWA olfactory neuron expressed a significantly greater proportional enrichment of upregulated GPCRs than expected if chemoreceptors genes were upregulated uniformly across

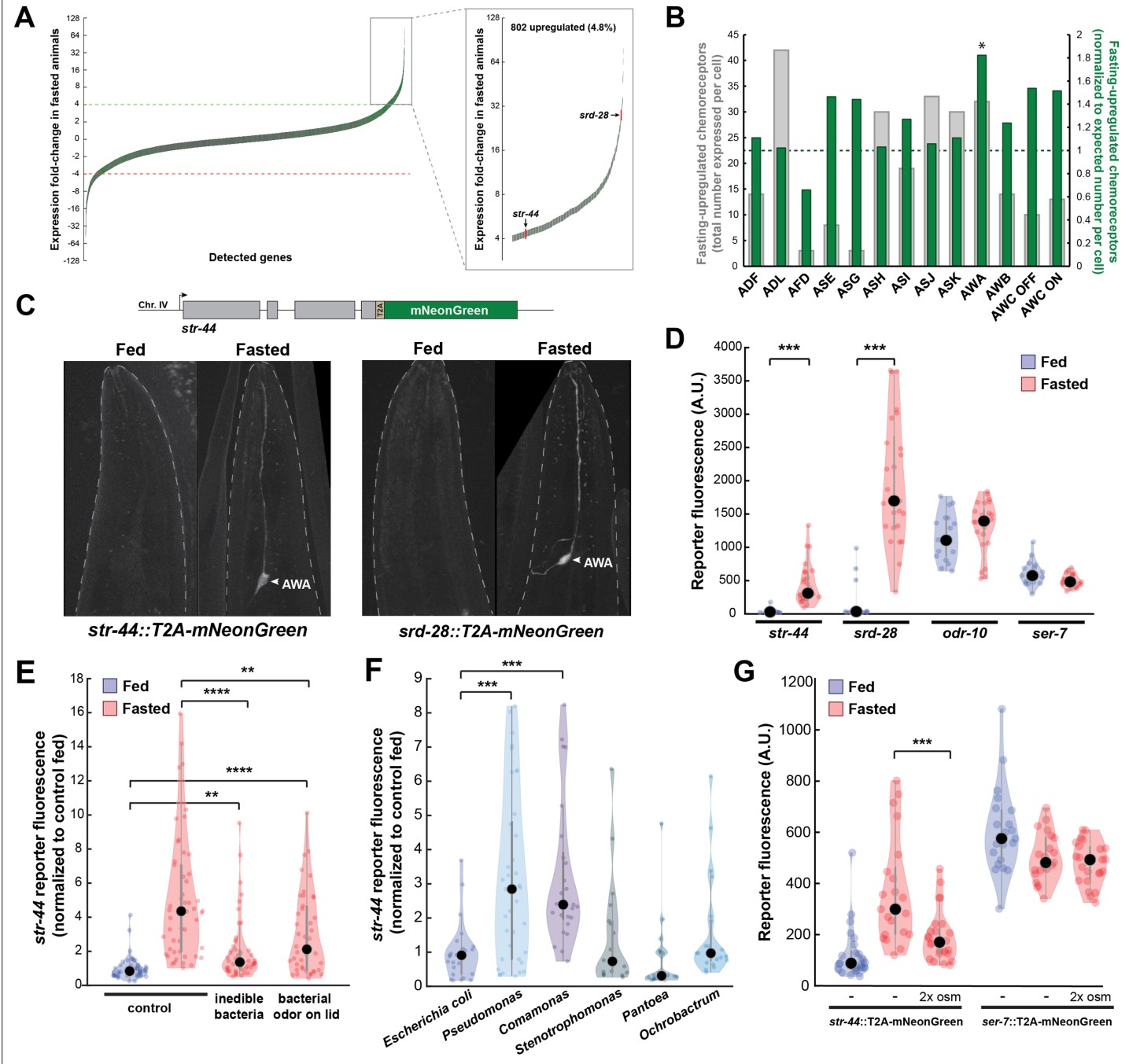

**Figure 1.** Diverse states and stimuli influence the expression of chemosensory G-protein-coupled receptors (GPCRs) in the AWA olfactory neuron. (**A**) Fold-change of transcript levels for all detected genes (n = 16,591) after 3 hr of fasting compared to well-fed condition. Dashed lines represent the fold-change cutoff for upregulated (green) or downregulated (red) genes. Green: chemoreceptors; gray: all other genes. Inset: zoomed view of upregulated genes. (**B**) Number of upregulated chemoreceptors in each of the 12 amphid neurons, based on the CeNGEN expression database. Gray bars: total number of upregulated chemoreceptor genes expressed per neuron. Dark green bars: fraction of upregulated chemoreceptors per neuron, divided by the fraction of all chemoreceptors expressed in that neuron (i.e., enrichment of fasting-upregulated chemoreceptors in each neuron, compared to number expected by chance based on total number of chemoreceptors in the neuron). *p<0.05, Bonferroni-corrected chi-squared test. (**C**) Top: endogenous CRISPR/Cas9 tagging strategy to visualize chemoreceptor gene expression in vivo, in which a T2A-mNeonGreen reporter sequence was inserted immediately before the stop codon. Bottom: example images of reporter strains in fed and fasted states. AWA neuron is indicated by arrowhead. (**D**) Quantification of chemoreceptor reporter gene fluorescence for two state-dependent chemoreceptors (*str-44* and *srd-28*), the known AWA chemoreceptor *odr-10*, and a control GPCR (*ser-7*) that is not upregulated by fasting. Note that there is no significant difference between fed and fasted for *odr-10* and *ser-7* reporters. (**E**) Relative expression of *str-44*::T2A-mNeonGreen in animals fed or fasted for 3 hr, animals

*Figure 1 continued on next page*

*Figure 1 continued*

exposed to inedible aztreonam-treated bacteria for 3 hr, and animals fasted in the presence of odors from an unreachable bacterial lawn (on lid of plate). (**F**) Relative expression *of str-44*::T2A-mNeonGreen in animals reared on *E. coli* (OP50) then fed on different bacteria for 3 hr. Bacterial strains used: OP50 (*E. coli*), PA14 (*Pseudomonas*), DA1877 (*Comamonas*), JUb19 (*Stenotrophomonas*), BIGb0393 (*Pantoea*), and Myb71 (*Ochrobactrum*). (**G**) Expression of *str-44*::T2A-mNeonGreen and *ser-7*::T2A-mNeonGreen (negative control) fed, fasted for 3 hr, or fasted on 300 mOsm plates (nematode growth medium [NGM] supplemented with an extra 150 mM sorbitol). Note that there is no significant effect of mild osmotic stress on the *ser-7* reporter. For (**D–G**), colored dots represent measurements from individual cells, black dots represent median values, and shaded area shows kernel density estimation for the data. Each condition measured in two independent sessions. \*\*\*p<0.001, \*\*p<0.01, \*p<0.05, two-tailed *t*-test with Bonferroni correction.

The online version of this article includes the following figure supplement(s) for figure 1:

**Figure supplement 1.** Identification of genes upregulated by fasting with pan-neural ribotagging.

neuron types (24/133 chemoreceptors, *Figure 1B*, green bars; *Figure 1—figure supplement 1D*). This suggests that AWA chemoreceptor expression levels are particularly sensitive to food deprivation.

To directly monitor the sites of chemoreceptor expression, we generated in vivo transcriptional reporters by inserting a t2a-mNeonGreen fluorescent reporter at the C-termini of several chemoreceptor genes via CRISPR-based gene editing (*Figure 1C*). As our ribotagging results suggested that these genes were upregulated after fasting, we compared mNeonGreen fluorescence for each of these chemoreceptors in fed and 3 hr fasted animals. For the *str-44* and *srd-28* chemoreceptor genes, we observed very little fluorescence in fed animals, but a significant increase specifically in AWA in fasted animals (*Figure 1D*; confirmation by co-expression of an AWA marker in *Figure 1—figure supplement 1E*). As a point of comparison, expression of a mNeonGreen reporter inserted into the well-characterized AWA-specific chemoreceptor *odr-10* was detectable in fed animals and displayed only a small, nonsignificant increase after fasting, consistent with the fact that it was only mildly upregulated by fasting in our ribotagging data (*Figure 1D*). As a negative control, we generated a reporter for the serotonin receptor *ser-7*, a GPCR in a different gene family whose expression was not altered in our ribotagging data, and observed that its expression was not affected by fasting. This suggests that the t2a-mNeonGreen transgene does not aberrantly confer fasting-dependent regulation. Overall, these reporter gene results provide a close match to our ribotagging data. These data suggest that fasting causes an upregulation of chemoreceptors in the AWA olfactory neuron.

Because the approximately fivefold increase in *str-44* expression upon food deprivation was particularly reliable and we were able to identify odorants that activate STR-44 (see below), we focused our experiments on this chemosensory GPCR. Depriving animals of their bacterial food for 3 hr impacts them in two ways: it leads to a change in metabolic state due to decreased ingestion, and it causes a change in sensory experience due to the removal of food sensory cues. To determine which of these effects influence *str-44* expression, we exposed animals to food cues under conditions where they were unable to ingest the food. First, we exposed animals to bacteria treated with aztreonam, which inhibits cell division and renders the bacteria too large to consume. This manipulation led to a level of *str-44* expression that was intermediate to well-fed and fasted animals (*Figure 1E*). This suggests that the ingestion of bacteria is necessary to fully suppress *str-44* expression, but also that non-nutritive components of the bacteria, such as volatile odorants, can partially suppress *str-44* expression. However, aztreonam treatment may also alter mechanosensory or chemical properties of the bacterial lawn, and therefore the sensory experience of the animal is not strictly identical to the untreated lawn. Therefore, we also used a second approach in which we exposed animals to food that was placed on the lid of the plate, rendering it inaccessible to the animal. As expected, this manipulation also produced a level of *str-44* expression that was intermediate to well-fed and fasted animals (*Figure 1E*). Together, these experiments suggest that both food sensory cues and the actual ingestion of food act to suppress olfactory receptor expression in AWA (*Figure 1E*).

In the wild, *C. elegans* interact with and ingest diverse microbial species (*Samuel et al., 2016*) that differ both in the odors that they emit and their metabolic contents. Given that AWA chemoreceptor expression is regulated by both sensory and metabolic cues, we hypothesized that AWA chemoreceptor expression might be modulated not just by the presence or absence of bacteria, but also by exposure to different bacterial food sources. To test this, after raising animals on the standard laboratory diet of *Escherichia coli* (OP50), we transferred them to plates where they were able to consume different bacterial species for 3 hr. We sampled representative bacterial strains from five different

genuses: *Stenotrophomonas* (JUb19), *Pantoea* (BIGb0393), *Comamonas* (DA1877), *Pseudomonas* (PA14), and *Ochrobactrum* (MYb71). Indeed, the levels of *str-44*::mNeonGreen reporter expression were increased significantly compared to *E. coli* OP50 controls when animals were exposed to *Comamonas* DA1877 or *Pseudomonas* PA14 (*Figure 1F*), two species that are naively attractive to *C. elegans* (*Shtonda and Avery, 2006*; *Zhang et al., 2005*). These changes could be due to differences in sensory cues and/or metabolic contents of these bacteria. By comparing the above results to conditions where DA1877 and PA14 were inaccessible on the lid of the plate, we found that the odors had differential effects on chemoreceptor expression. Relative to fasted controls, PA14 odor increased *str-44* reporter expression, whereas DA1877 odor suppressed *str-44* expression. Thus, the effect of PA14 exposure may be driven by volatile odors while the effect of DA1877 exposure may be driven by ingestion and/or physical contact with the bacteria (*Figure 1—figure supplement 1F*). Together, these data suggest that exposure to different bacterial odors and metabolic contents impacts the expression of AWA olfactory receptors.

We next sought to determine whether AWA chemoreceptor expression is exclusively controlled by feeding-related signals or, alternatively, whether it is impacted by a broader set of external and internal cues. Thus, we also examined whether the addition of aversive/stressful stimuli influenced *str-44* expression. We chose to use a mild osmotic stressor (300 mOsm growth media versus 150 mOsm in normal media) that modulates *C. elegans* behavior, but does not adversely impact animal growth rates or viability (*Yu et al., 2017*; *Zhang et al., 2008*). We found that animals fasted in the presence of this physiological stressor displayed *str-44* expression that was significantly suppressed relative to fasted controls (*Figure 1G*). However, the *ser-7*::mNeonGreen control reporter was unaffected by this mild osmotic stressor, indicating that these effects are not due to generic downregulation of GPCRs or diminished expression of mNeonGreen in response to osmotic stress (*Figure 1G*). Taken together, these results suggest that rather than relying on food-related signals alone, diverse external and internal cues converge on AWA to coordinately regulate the expression of the *str-44* chemoreceptor. Given that our ribotagging analysis identified >20 putative AWA chemoreceptors impacted by fasting, it is possible that many AWA chemoreceptors display similar gene expression changes in response to various states.

## The state-dependent chemoreceptor STR-44 acts in AWA to detect the attractive odors propyl acetate and butyl acetate

We next sought to understand how convergent signaling onto AWA might allow animals to generate context-appropriate sensorimotor responses. Therefore, we focused on examining how the state-dependent AWA chemoreceptors influence *C. elegans* sensorimotor behaviors. AWA is an olfactory neuron that drives attraction to volatile odors (*Ferkey et al., 2021*). To identify odors that activate the state-dependent chemoreceptors, we generated strains that ectopically expressed either *str-44* or *srd-28* in the nociceptive sensory neuron ASH and asked whether this could confer repulsion to odors previously shown to activate AWA (*Larsch et al., 2013*). To reduce native responses to these odors, we performed these experiments in a genetic background with an *odr-7* mutation, which inactivates AWA, and a *tax-4* mutation, which prevents sensory transduction in other olfactory neurons. We tested olfactory behavior using a chemotaxis assay, measuring movement toward or away from each of the tested odors. Due to their genetic background, we predicted that these animals would generally have neutral responses to the tested odors. However, if an odor were a ligand for the *str-44* or *srd-28* receptor, then expression of that receptor in ASH should drive a repulsive response to the odor. As expected, the ASH::*str-44* and ASH::*srd-28* strains had neutral responses to most tested odors, indistinguishable from the *odr-7;tax-4* control strain. However, the ASH::*str-44* strain was significantly repulsed by two structurally similar esters, propyl acetate and butyl acetate, suggesting that *str-44* chemoreceptor expression in ASH is sufficient to confer detection of these two odors (*Figure 2A*). The ASH::*srd-28* strain did not generate any significant responses to the odors tested, suggesting that *srd-28* may detect other odors that we did not test here (*Figure 2A*). These data suggest that the odors propyl and butyl acetate are detected by the *str-44* olfactory receptor.

We next examined how wild-type animals respond to propyl and butyl acetate. Wild-type animals were strongly attracted to both of these odors. However, *odr-7* mutants lacking AWA had significantly decreased responses to these odors, indicating that AWA is necessary for navigation to propyl and butyl acetate (*Figure 2B*). Based on these results, we examined whether AWA calcium responses to

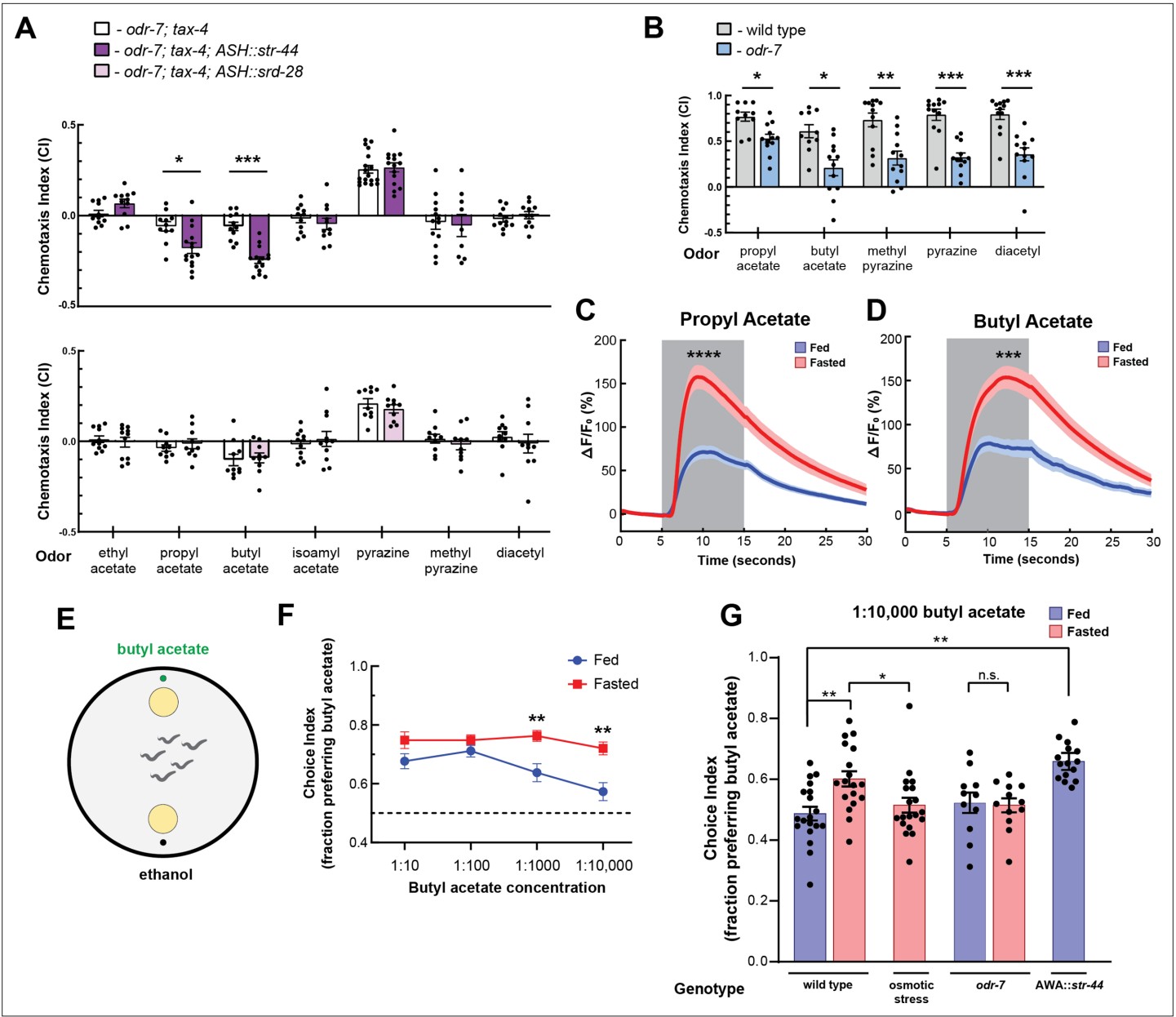

**Figure 2.** The state-dependent chemoreceptor *str-44* responds to appetitive odors and controls state-dependent odor preference. (**A**) Chemotaxis behavior in *odr-7(ky4);tax-4(p678)* animals, and *odr-7;tax-4* animals expressing either *str-44* or *srd-28* in the nociceptive neuron ASH (*sra-6p*). Bars show mean ± SEM. n = 2–4 days, each with 3–6 assay plates per odor, and 50–200 animals per plate (*p<0.01, ***p<0.001, *t*-test with Bonferroni correction). (**B**) Chemotaxis behavior to odors of interest in wild-type and *odr-7* mutants, which lack a functional AWA neuron. Bars show mean ± SEM. n = 2 days, each with 5–7 assay plates per odor (*p<0.01, **p<0.005, ***p<0.001, *t*-test with Bonferroni correction). *odr-7* mutant chemotaxis to a wide range of other odors is not impaired (*Sengupta et al., 1994*). (**C**) AWA calcium imaging in response to a 10 s addition of $10^{-6}$ propyl acetate in fed (blue) and 3 hr fasted (red) worms. n = 31 fed animals, 13 fasted animals, two trials per animal. Plots show mean ± SEM. ****p<0.0001, *t*-test with Bonferroni correction. (**D**) AWA calcium imaging in response to 10 s addition of $10^{-6}$ butyl acetate in fed (blue) and 3 hr fasted (red) worms. n = 20 fed animals, 14 fasted animals, two trials per animal. Plots show mean ± SEM. ***p<0.001, *t*-test with Bonferroni correction. (**E**) Schematic of food choice assay. Animals are placed equidistant from two *E. coli* OP50 lawns, one with an adjacent drop of butyl acetate and one with an adjacent drop of ethanol (its dilutant). (**F**) Food choice behavior of fed and fasted animals, showing fraction of animals that chose the food lawn with a spot of butyl acetate at the indicated concentration. Plots show mean ± SEM. n = 9–20 plates per odor over two independent sessions, each with 40–200 animals. **p<0.01, *t*-test with Bonferroni correction. (**G**) Food choice behavior of fed and fasted WT or mutant animals, animals experiencing osmotic stress during fasting, and fed animals overexpressing *str-44* in AWA (*gpa-6p*), showing fraction of animals that chose the food lawn with a spot of 1:10,000 butyl acetate. Bars show mean ± SEM. n > 12 plates per odor over two independent sessions, each with 40–200 animals. ****p<0.0001, **p<0.01, *t*-test with Bonferroni correction. Note that there is no significant effect between fed and fasted for *odr-7* mutants.

*Figure 2 continued on next page*

*Figure 2 continued*

The online version of this article includes the following figure supplement(s) for figure 2:

**Figure supplement 1.** AWA calcium responses to butyl acetate and propyl acetate are potentiated by fasting, and state-dependent odor choice depends on AWA chemoreceptors.

butyl and propyl acetate were state-dependent. We measured AWA GCaMP signals while delivering 10 s pulses of odor via microfluidic delivery. Consistent with prior work (*Larsch et al., 2013*), AWA calcium levels increased in response to the addition of either butyl or propyl acetate (*Figure 2C and D*, *Figure 2—figure supplement 1*). Notably, we found that fasted animals exhibited significantly increased responses to these odors compared to well-fed animals (*Figure 2C and D*, *Figure 2—figure supplement 1*). Thus, AWA calcium responses to the cues detected by the *str-44* chemoreceptor are potentiated in fasted animals, when *str-44* is expressed at high levels. Because AWA can detect butyl and propyl acetate in the fed state, when *str-44* levels are often undetectable, we expect that multiple chemoreceptors contribute to detection of these odorants. In addition, we have not ruled out that other fasting-upregulated chemoreceptors in AWA (*Figure 1B*) may also contribute to the fasting-induced potentiation of the response to propyl and butyl acetate.

## STR-44 expression drives state-dependent enhancement of behavioral preference for the attractive odor butyl acetate

We hypothesized that animals' behavioral responses to *str-44*-sensed odors would be modulated by the states and stimuli that alter *str-44* expression levels, and that direct perturbations of *str-44* expression would also impact behavior. To test this, we used a modified food choice assay (*Worthy et al., 2018*) in which animals choose between two small and equidistant lawns of *E. coli* OP50 bacterial food. We placed a spot of butyl acetate adjacent to one lawn, and as a control, placed a spot of ethanol adjacent to the other lawn (*Figure 2E*). We chose to use this assay instead of chemotaxis because fasted animals display generically reduced movement in the presence of single odorants but display robust movement in food choice assays. At high concentrations of odor, both fed and fasted animals were attracted to the food lawn with butyl acetate (*Figure 2F*). However, at lower odor concentrations, fasted animals were significantly more likely than fed animals to approach the food lawn with butyl acetate (*Figure 2F*). This suggests that fasted animals display increased sensitivity to the attractive odor butyl acetate, consistent with the increased expression of the butyl acetate receptor *str-44* in fasted animals. To test whether increased expression of *str-44* could directly drive increased sensitivity to butyl acetate, we generated a strain that overexpresses *str-44* under a constitutive AWA promoter, driving overexpression in both fed and fasted animals. Indeed, well-fed animals from this strain were significantly more likely to approach the butyl acetate food lawn compared to well-fed wild-type animals, phenocopying the fasted state (*Figure 2G*). We also tested whether the fasting-induced increase in butyl acetate sensitivity requires AWA and/or *str-44*. *odr-7* mutants lacking a functional AWA did not display a fasting-induced increase in butyl acetate attraction, suggesting that AWA is required for this effect (*Figure 2G*). In addition, *str-44;srd-28* double mutants displayed an attenuated behavioral response where their attraction to the butyl acetate lawn was similar in fed and fasted conditions, suggesting that *str-44* and/or *srd-28* are necessary for normal fasting-induced enhancement of butyl acetate preference (*Figure 2—figure supplement 1D*). Together, these results indicate that *C. elegans* displays an AWA-dependent increase in sensitivity to butyl acetate upon fasting and that expression of the fasting-upregulated AWA chemoreceptor *str-44* can drive increased sensitivity to this odor.

We next examined whether other manipulations that increase or decrease *str-44* expression could likewise modify butyl acetate sensitivity. Since the presence of mild osmotic stress blocks the upregulation of *str-44* in AWA during fasting, we examined whether this manipulation also blocks the fasting-induced increase in butyl acetate sensitivity. Indeed, when we tested fasted animals undergoing osmotic stress, they were no more likely to approach the butyl acetate food lawn than well-fed controls (*Figure 2G*). Together with the direct manipulations of *str-44* levels described above, these results suggest that the level of *str-44* expression in AWA, which is set via the integration of multiple states, drives butyl acetate sensitivity. More broadly, these results are consistent with the notion that integrated state-dependent changes in the expression of *str-44* and potentially other chemoreceptors

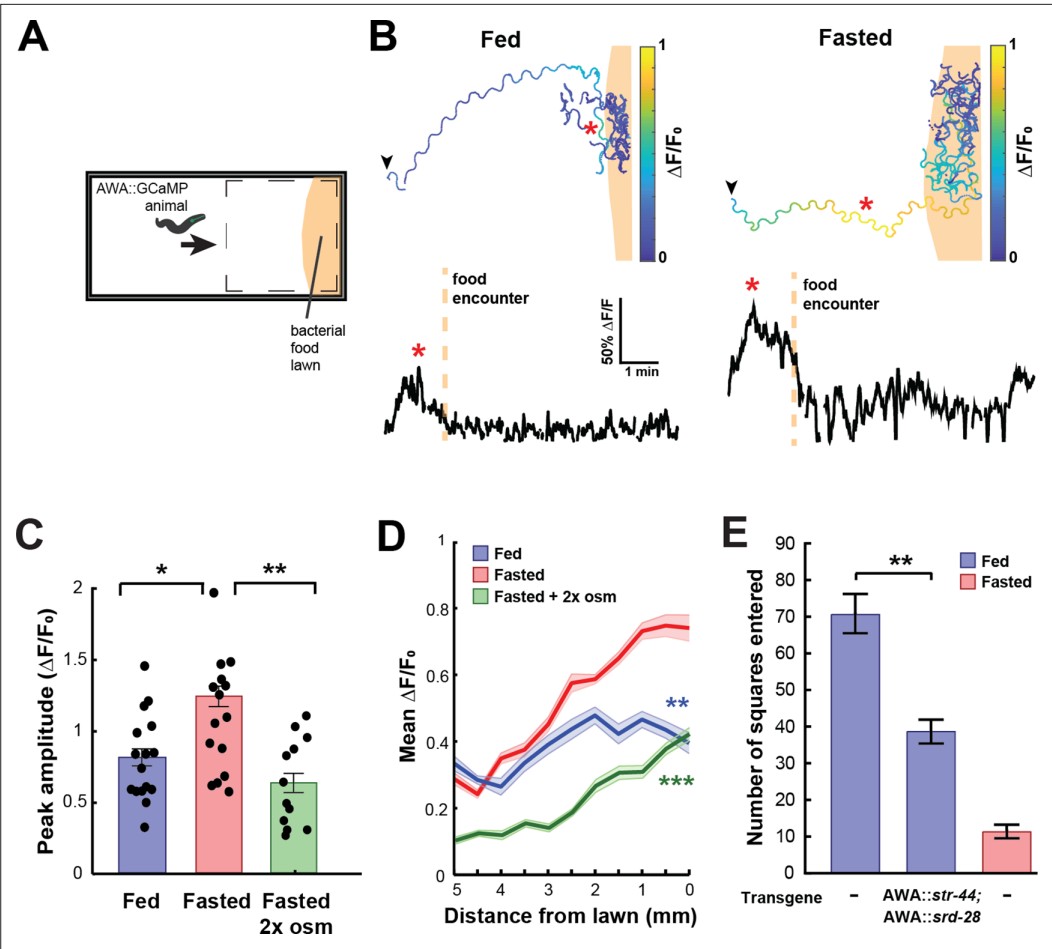

**Figure 3.** State-dependent AWA calcium responses and behavioral responses to bacterial food. (**A**) Schematic of freely moving calcium imaging assay. Animals expressing AWA::GCaMP2.2b are picked to an agar pad and allowed to freely navigate to an *E. coli* OP50 lawn. Dashed box indicates field of view of the microscope. (**B**) Example AWA GCaMP recording of an individual fed (left) or fasted (right) animal. Top: movement trajectories of animals, with colors indicating AWA GCaMP fluorescence. Bottom: GCaMP trace for the same animals. Red asterisks: peak of GCaMP signal. Dashed line: time of food encounter. (**C**) Amplitude of each GCaMP peak in fed animals, fasted animals, and animals fasted in the presence of osmotic stress (300 mOsm). Bars show mean ± SEM. *p<0.05, **p<0.01, *t*-test with Bonferroni correction. (**D**) Mean GCaMP signal in fed or fasted animals binned by the animal's distance from the lawn boundary (0.5 mm bins). n = 18 animals per condition. Plots show mean ± SEM. Two-way ANOVA, significant effect of feeding state (***p<0.001, **p<0.01). (**E**) On-food exploration assay comparing wild-type fed and fasted animals with fed animals overexpressing *str-44* and *srd-28* chemoreceptors in AWA (*gpa-6p*). n = 15–30 animals per condition. Plots show mean ± SEM. **p<0.01, *t*-test.

The online version of this article includes the following figure supplement(s) for figure 3:

**Figure supplement 1.** AWA GCaMP peak duration and locomotor controls.

may alter sensory responses to specific odors, allowing animals to modulate their navigation based on the sum of their recent experience and physiology.

## AWA responses to food are influenced by internal state, and the state-dependent olfactory receptors drive enhanced responses to food

In natural environments, *C. elegans* encounter complex mixtures of olfactory stimuli rather than mono-molecular odorants, so we next asked whether the response of AWA to bacterial odors is influenced by the same states that modify AWA chemoreceptor expression. Previous studies have shown that AWA responds to bacterial volatiles (*Zaslaver et al., 2015*), which are complex mixtures of hetero-geneous odorants, including esters like butyl and propyl acetate. To examine state-dependent AWA

calcium responses to food odor gradients, we performed AWA GCaMP imaging in freely moving fed or fasted animals as they navigated toward a lawn of *E. coli* OP50 food (*Figure 3A*). Across conditions, AWA displayed notable calcium peaks as animals navigated toward the food lawn (*Figure 3B*). These responses mostly occurred when animals were in close proximity to the food lawn and when they were moving up the odor gradient toward the food. The amplitudes and durations of these peaks were significantly increased in fasted animals compared to fed animals (*Figure 3C*, *Figure 3—figure supplement 1A*). This resulted in a significant difference in overall AWA activity between fed and fasted animals that was maximally apparent shortly before lawn encounter (*Figure 3D*). The fasting-induced increase in AWA food responses was attenuated by exposing animals to mild osmotic stress during fasting (*Figure 3C and D*), matching the above results that this stressor suppresses the fasting-induced increase in *str-44* expression. These results indicate that AWA responses to bacterial food odor gradients are influenced by an integrated internal state including fasting and stress.

These experiments suggested that changes in AWA chemoreceptor expression might influence food-driven behaviors, much like they influence butyl acetate odor preference. Thus, we utilized an assay of bacterial food exploration during foraging. Fasted animals reduce their locomotion on a food lawn compared to fed animals, reflecting increased exploitation of a food source after fasting (*Ben Arous et al., 2009*; *Shtonda and Avery, 2006*). We hypothesized that fed animals that overexpress *str-44* in AWA would behave as if they were fasted, much like in the above food choice experiments. Indeed, we observed that overexpressing *str-44* and *srd-28* in AWA led to a marked decrease in exploration in the fed state, mimicking the behavior of fasted wild-type animals (*Figure 3E*). This effect on movement was only observed in the presence of bacterial food as the same overexpression strain displayed wild-type speed in the absence of food (*Figure 3—figure supplement 1B*). Thus, increasing *str-44* and *srd-28* expression in AWA is sufficient to alter food-driven changes in locomotion, partially mimicking the behavior of fasted animals. Taken together with the above results, these data suggest that AWA calcium responses to bacterial food are enhanced by fasting, and that increased expression of the fasting-induced olfactory receptor *str-44* results in fasted-like behavioral responses to food. Overall, the behavioral studies that we have carried out indicate that state-dependent modulation of AWA chemoreceptor expression, in particular *str-44* expression, alters the animal's sensorimotor behaviors related to food navigation and foraging.

## Signaling from a set of food sensory neurons to AWA regulates *str-44* expression

We next sought to understand the mechanisms by which AWA integrates diverse external and internal cues to influence sensorimotor behaviors. Our overall approach was to first determine the molecular and neural pathways that convey each sensory stimulus or state to AWA and then examine how they interact. We first examined how food sensory cues influence AWA olfactory receptor expression. We examined *str-44*::mNeonGreen reporter fluorescence in *tax-4* mutants that have defective sensory transduction in many food-responsive sensory neurons (*Ferkey et al., 2021*). Importantly, AWA sensory transduction does not require *tax-4* (*Ferkey et al., 2021*). Well-fed *tax-4* mutants exhibited a striking increase in *str-44* expression compared to wild-type controls, with expression levels even greater than wild-type fasted animals (*Figure 4A*). Correspondingly, well-fed *tax-4* mutants exhibited increased attraction to the STR-44-sensed odor butyl acetate (*Figure 4F*). A potential concern is that an increase in *str-44* expression may occur if a mutation reduces food intake; however, we found that *tax-4* animals displayed normal feeding rates (*Figure 4—figure supplement 1A* shows normal feeding rates for *tax-4* and all other mutants/transgenics with elevated *str-44* expression in fed animals). This phenotype could be rescued by expressing the *tax-4* cDNA under its own promoter (*Figure 4A*). This suggests that impaired sensory transduction in one or more *tax-4*-expressing sensory neuron increases *str-44* expression in the AWA sensory neuron. In addition, we found that *tax-4* mutants also have elevated expression of another AWA chemoreceptor, *srd-28* (*Figure 4—figure supplement 1G*). Thus, crosstalk between sensory neurons regulates AWA chemoreceptor expression. Such crosstalk could occur either through direct synaptic communication between sensory neurons, extrasynaptic neuromodulation, or feedback through bidirectional interneurons.

We next determined which *tax-4*-expressing neurons functionally regulate *str-44* expression. To do so, we impaired the function of these neurons individually and examined the effect on *str-44*::mNeonGreen reporter fluorescence. Chemogenetic silencing of AWB, BAG, ASI, AQR/PQR/URX,

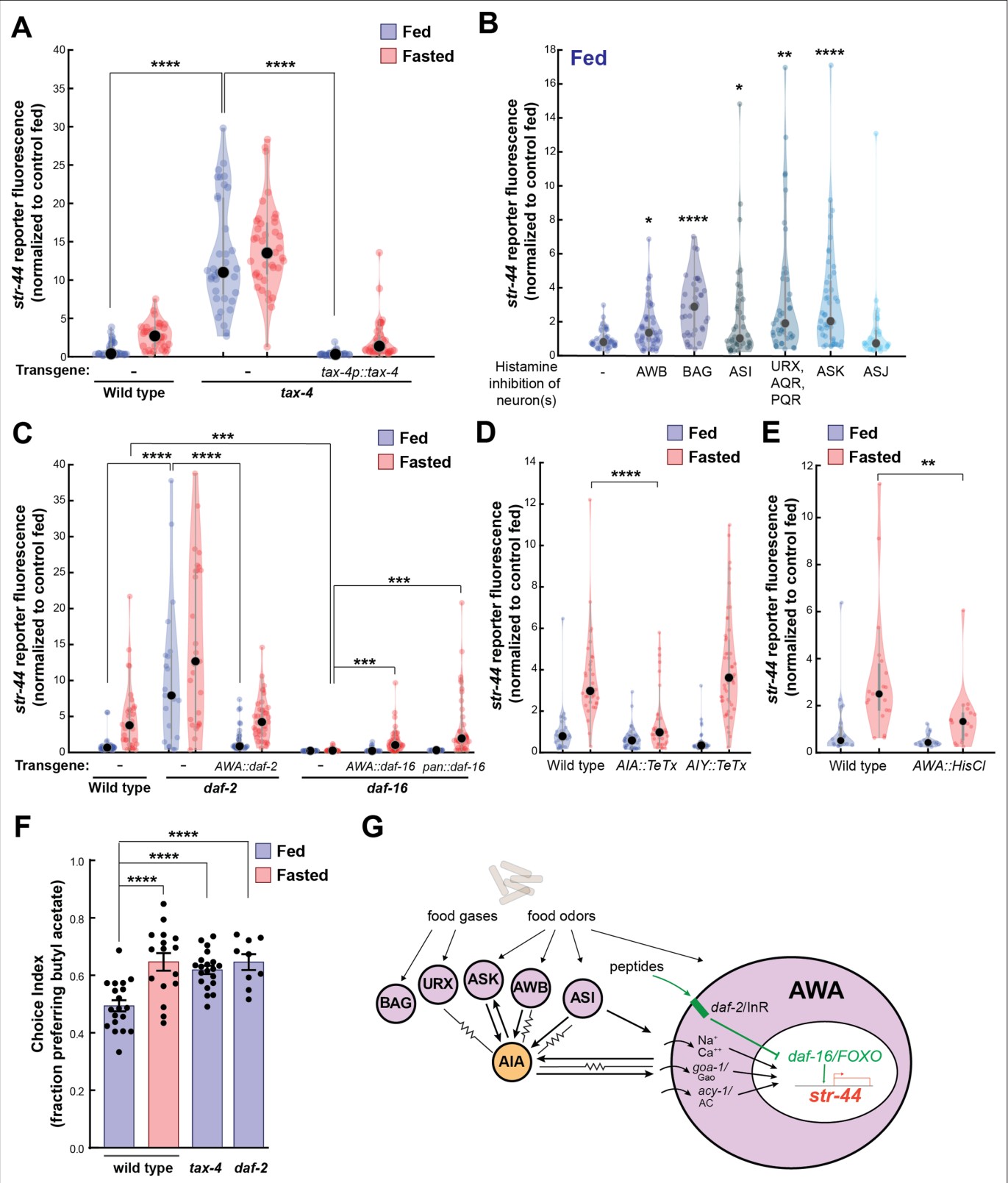

**Figure 4.** Signaling from food sensory neurons to AWA modulates *str-44* expression via multiple pathways. (**A**) Relative expression of *str-44*::T2A-mNeonGreen in *tax-4(p678)* mutants with and without *tax-4::tax-4* rescue construct, compared to wild-type. ****p<0.0001, two-tailed *t*-test with Bonferroni correction. (**B**) *str-44*::T2A-mNeonGreen expression following 3 hr histamine-induced inhibition of sensory neurons while feeding. Histamine-gated chloride channel (HisCl1) transgenes expressed with cell-specific promoters: AWB (*str-1*p::HisCl1), BAG (*gcy-33*p::HisCl1), ASI (*srg-47*p::HisCl1),

*Figure 4 continued on next page*

*Figure 4 continued*

URX, AQR, PQR (*gcy-36*p::HisCl), ASK (*sra-9*p::HisCl1), and ASJ (*srh-11*p::HisCl1). *p<0.05, **p<0.01, ***p<0.001, ****p<0.0001, two-tailed *t*-test with Bonferroni correction. Note that there is no significant effect of ASJ silencing compared to controls. (**C**) Relative expression of *str-44*::T2A-mNeonGreen in *daf-2(m41)*, and *daf-16(mu86)* mutants and mutants bearing transgenes for AWA-specific (*gpa-6*p) or pan-body (*dpy-30*p) rescues for *daf-2* and *daf-16*. ***p<0.001, ****p<0.0001, two-tailed *t*-test with Bonferroni correction. (**D**) Relative expression of *str-44*::T2A-mNeonGreen in animals with synaptic silencing of AIA (*gcy-28.d*p::TeTx) or AIY (*ttx-3*p::TeTx). ****p<0.0001, two-tailed *t*-test with Bonferroni correction. Note that there is no significant effect of AIY synaptic silencing, compared to controls. (**E**) Relative expression of *str-44*::T2A-mNeonGreen following 3 hr histamine-induced inhibition of AWA (*gpa-6*p::HisCl1). **p<0.01, two-tailed *t*-test with Bonferroni correction. (**F**) Food choice behavior in wild-type, *tax-4*, or *daf-2* mutant animals showing fraction of animals that chose the food lawn with a spot of 1:10,000 butyl acetate. Bars show mean ± SEM. n > 15 plates per odor over two independent sessions, each with 40–200 animals. ****p<0.0001, *t*-test with Bonferroni correction. (**G**) Model depicting the pathways that signal from food sensory neurons to AWA. Signaling occurs via insulin signaling, modulation of AWA activity, and modulation of AWA G protein signaling. For (**A–E**), colored dots represent measurements from individual cells, black dots represent median values, and shaded area shows kernel density estimation for the data. Each condition measured in two independent sessions.

The online version of this article includes the following figure supplement(s) for figure 4:

**Figure supplement 1.** Controls and additional mutants for single-neuron inhibition experiments, and *srd-28* expression changes in relevant mutants.

**Figure supplement 2.** Manipulations of AWA activity, but not *str-44* ligand exposure or AWA synaptic transmission, modify *str-44* expression.

or ASK via a histamine-gated chloride channel (HisCl) (*Pokala et al., 2014*) led to a significant increase in *str-44* expression in well-fed animals, whereas ASJ silencing had no effect (*Figure 4B*; feeding controls in *Figure 4—figure supplement 1B*; histamine controls in *Figure 4—figure supplement 1C*). In fasted animals, chemogenetic silencing did not further increase *str-44* expression except in ASK, and silencing of ASI reduced *str-44* expression relative to fasted controls (*Figure 4—figure supplement 1D and E*), which may reflect distinct requirements for ASI in fed versus fasted states. Genetic ablation of AWC and ASE via *ceh-36* mutation also had no effect on *str-44* expression in fed animals (*Figure 4—figure supplement 1F*; see also for additional mutants and transgenics that corroborate these findings). These data indicate that a defined set of *tax-4*-expressing neurons inhibit AWA *str-44* expression in well-fed animals. Notably, each of the neurons that inhibit *str-44* expression have been previously shown to detect food sensory cues (*Ferkey et al., 2021*). Taken together, these data suggest that food sensory cues are detected by AWB, BAG, ASI, AQR/PQR/URX, and ASK, which signal to AWA to inhibit *str-44* expression while animals are feeding.

In principle, these food sensory neurons could signal to AWA through neuropeptide release and/or synaptic outputs onto downstream circuit components that in turn synapse onto AWA. We separately examined these possibilities. Among the neuropeptidergic pathways, the insulin/IGF-1 signaling pathway (IIS) has been most prominently linked to feeding state, and several of the food sensory neurons that regulate *str-44*, such as ASI, release insulin peptides. Insulin acts primarily through the *daf-2* insulin receptor (*Murphy and Hu, 2013*). Activation of *daf-2* alters gene expression by inhibiting the *daf-16*/FOXO transcription factor, which is also a target of other cellular signaling pathways (*Landis and Murphy, 2010*). Loss of the *daf-2* insulin receptor also caused a strong de-repression of *str-44* and a corresponding increase in butyl acetate attraction (*Figure 4C and F*). Conversely, *daf-16* mutants showed nearly abolished *str-44* expression (*Figure 4C*). The *daf-2* insulin receptor and *daf-16*/FOXO are broadly expressed, so we next examined where they function. Expression of the *daf-2* cDNA specifically in AWA rescued *str-44* expression to wild-type levels (*Figure 4C*). Expression of the *daf-16a* cDNA specifically in AWA rescued *str-44* expression, albeit not completely; expression of this cDNA in all tissues yielded the same result, suggesting that additional isoforms of *daf-16* are required (*Figure 4C*). Finally, we determined that *daf-2* and *daf-16* mutants respectively increase and decrease *srd-28* expression in AWA (*Figure 4—figure supplement 1H*). Together, these results suggest that *daf-2* and *daf-16* act in AWA to regulate olfactory receptor expression.

We also examined whether other neuropeptide signaling pathways could influence *str-44* expression. Most neuropeptides act on GPCRs that couple to the G proteins Gαo/*goa-1*, Gαs/*gsa-1*, or Gαq/*egl-30*, which are all natively expressed in AWA (*Taylor et al., 2021*). Thus, we mimicked the activation of neuropeptide receptors in AWA via AWA-specific expression of constitutively active versions of these G proteins and examined the impact on *str-44* expression. AWA-specific expression of Gαo/*goa-1(gf)* or *acy-1(gf)*, a key Gαs/*gsa-1* effector, drove a robust increase in *str-44* expression, whereas expression of Gαq/*egl-30(gf)* had only a mild effect (*Figure 4—figure supplement 2A*). This suggests that activation of specific G protein signaling pathways in AWA can influence *str-44*

expression and raises the possibility that additional neuropeptides that act through these canonical pathways regulate *str-44*.

We next tested whether sensory habituation in AWA is a suitable explanation for the odor-driven reduction in *str-44* expression. We exposed wild-type animals to the *str-44* odorants butyl acetate or propyl acetate during 3 hr of either feeding or fasting and measured *str-44* expression. If habituation explains this effect, we would expect that exposure to these odorants would reduce *str-44* expression in fasted animals. However, we observed no differences between odor-exposed animals and controls (*Figure 4—figure supplement 2B*), suggesting that *str-44* expression levels are not modulated by activation of the STR-44 receptor.

Finally, we examined whether the food sensory neurons could feasibly signal to AWA via action on downstream neural circuits. AWA receives strong synaptic input (>3 synapses) from three neurons in the *C. elegans* wiring diagram: ASI, one of the food sensory neurons that represses *str-44* expression; and AIA and AIY, which are second-order neurons in the chemosensory circuit that together receive synaptic inputs from all five of the food sensory neurons that inhibit *str-44* (*White et al., 1986*; *Witvliet et al., 2021*). To test whether AIA and AIY are required for proper *str-44* regulation, we inactivated these cells via expression of tetanus toxin light chain (TeTx). Synaptic silencing of AIA led to a significant decrease in *str-44* expression in fasted animals, whereas synaptic silencing of AIY had no effect (*Figure 4D*). In addition, direct chemogenetic silencing of AWA also inhibited *str-44* expression, suggesting that modulating AWA activity itself can influence *str-44* expression (*Figure 4E*; histamine controls in *Figure 4—figure supplement 2C*). To distinguish between a direct effect of AWA activity on intracellular gene expression and a feedback effect of reduced AWA synaptic transmission, we also measured *str-44* expression in animals expressing TeTx in AWA. Inhibition of synaptic release from AWA had no effect on *str-44* expression, suggesting that AWA activity autonomously controls AWA chemoreceptor levels (*Figure 4—figure supplement 2D*). These data suggest that a sensory circuit consisting of several food sensory neurons and their downstream synaptic target AIA regulates AWA chemoreceptor expression. Taken together with the above results, these data reveal that several signaling mechanisms – insulin signaling, G-protein signaling, and activity-dependent signaling – allow a defined set of food sensory neurons to regulate olfactory receptor expression in AWA (*Figure 4G*).

## *rict-1*/TORC2 signaling in the intestine signals to AWA to underlie metabolic regulation of AWA chemoreceptor expression

In addition to food sensory signals, our experiments suggest that the actual ingestion of food influences chemoreceptor expression in AWA (*Figure 1E and F*). We therefore sought to identify pathways that link physiological fasting to *str-44* expression levels in AWA. Monoaminergic neuromodulators including serotonin and octopamine can act as internal signals of food availability (*Rhoades et al., 2019*; *Sawin et al., 2000*; *Srinivasan et al., 2008*), so we examined *cat-1*/VMAT mutants, which are defective in the release of these neuromodulators. However, these mutants displayed normal *str-44* expression in fed and fasted states (*Figure 5A*). Likewise, *pdfr-1* animals lacking PDF neuropeptide signaling, which acts in opposition to serotonin (*Flavell et al., 2013*), displayed normal *str-44* expression (*Figure 5A*). We next examined whether changes in internal fat stores, which can impact nervous system function (*Witham et al., 2016*), influence *str-44* expression. However, mutations that disrupt the ability of the animal to store fat (*mxl-3,* MAX transcription factor) or metabolize triglycerides (*atgl-1*, adipose triglyceride lipase) did not alter *str-44* expression (*Figure 5A*). We also examined other nutrient signaling pathways. For example, *aak-1;aak-2* animals lacking the nutrient sensor AMP kinase (AMPK) displayed normal *str-44* expression levels (*Figure 5A*). Thus, *str-44* expression is not responsive to biogenic amines, internal fat stores, or AMPK signaling.

We next tested components of the TOR pathway, another crucial nutrient sensor and regulator of metabolic processes. Loss of the essential TORC1 complex component *raga-1*/RagA modestly reduced fasted *str-44* expression (*Figure 5B*). In contrast to the inhibitory pathways engaged by the presence of food and food odor, *raga-1* appears to be involved in a positively acting pathway engaged by the absence of food. In addition, loss of the essential TORC2 component *rict-1*/Rictor led to a robust increase in *str-44* expression in fed and fasted animals (*Figure 5B*) and a corresponding behavioral phenotype, increasing butyl acetate attraction (*Figure 5D*). *rict-1* is broadly expressed, but several of its metabolic functions require expression in the intestine (*Soukas et al., 2009*). Indeed, we found that the elevated *str-44* expression in *rict-1* mutants was fully rescued by expression of a *rict-1*

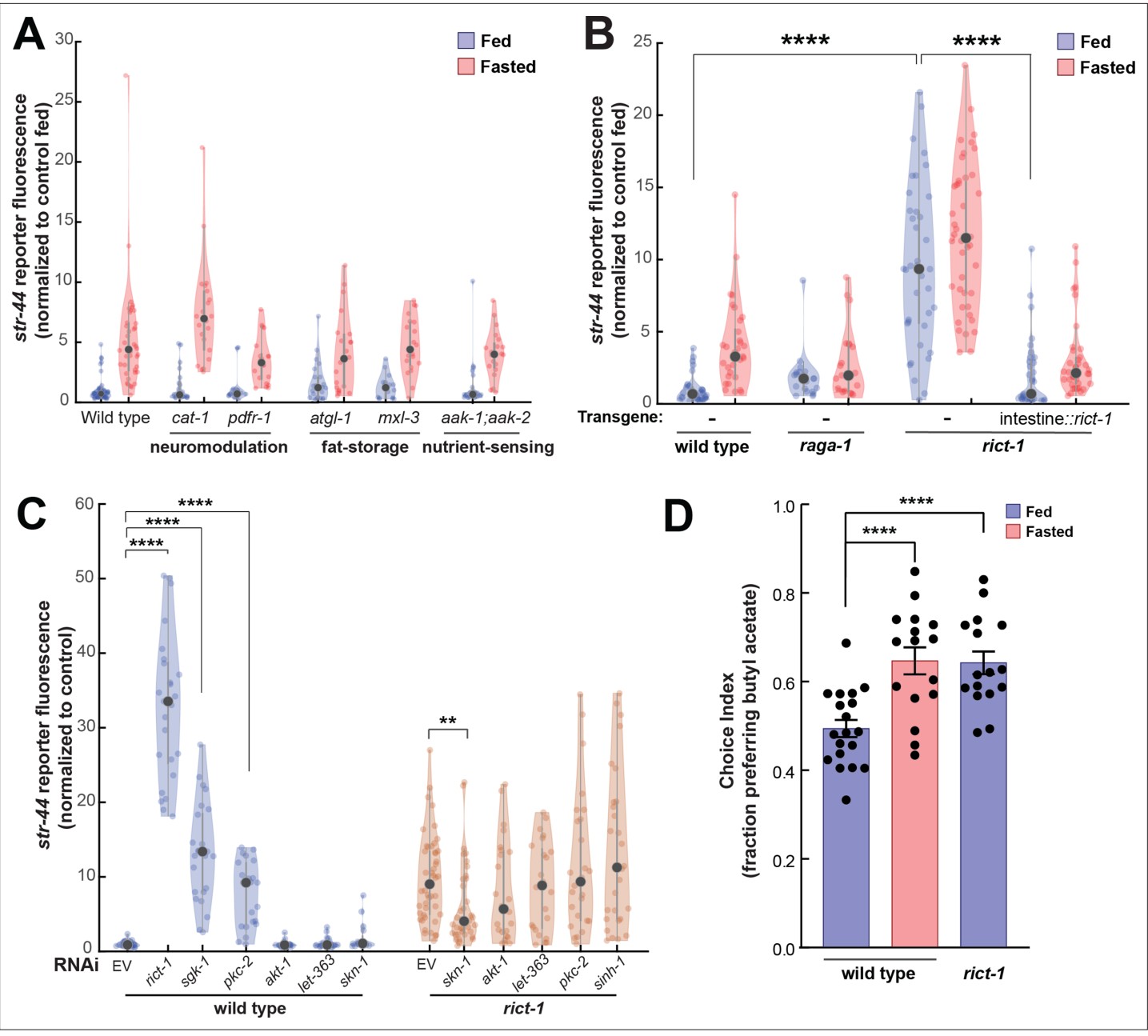

**Figure 5.** TORC2 signaling from the gut controls AWA *str-44* expression and odor preference. (**A**) Relative expression of *str-44*::T2A-mNeonGreen in *cat-1(e1111)*, *pdfr-1(ok3425)*, *atgl-1(gk176565)*, *mxl-3(ok1947)*, *aak-1(tm1944);aak-2(gt33)* mutants compared to wild-type. All mutant genotypes shown are statistically indistinguishable from the corresponding wild-type controls, and all fed versus starved comparisons within each genotype are significant at p<0.001 (two-tailed *t*-test). (**B**) Relative expression of *str-44*::T2A-mNeonGreen in *raga-1(ok386)* and *rict-1(ft7)* mutants and *rict-1* mutant bearing transgene for intestinal-specific (*ges-1p*) rescue for *rict-1(ft-7)* compared to wild-type when fed and fasted. *raga-1(ok386)* fed versus starved comparison is not significant by two-tailed *t*-test (p>0.05). (**C**) Relative expression of *str-44*::T2A-mNeonGreen in animals fed on *E. coli* HT115 with empty RNAi vector, *rict-1(RNAi)*, *sgk-1(RNAi)*, *pkc-2(RNAi)*, *akt-1(RNAi)*, *let-363(RNAi)*, *skn-1(RNAi)*, or *sinh-1(RNAi)*. Left: wild-type animals fed on RNAi. Right: *rict-1(ft7)* mutant animals fed on RNAi to identify suppressors. Unless otherwise marked, RNAi of genes resulted in no significant difference from controls. (**D**) Food choice behavior in wild-type or *rict-1* mutant animals showing fraction of animals that chose the food lawn with a spot of 1:10,000 butyl acetate. Bars show mean ± SEM. n > 15 plates per odor over two independent sessions, each with 40–200 animals. ****p<0.0001, *t*-test with Bonferroni correction. For (**A–C**), each condition measured in two independent sessions. ****p<0.0001, **p<0.01by two-tailed *t*-test. Colored dots represent from individual cells, black dots represent median values, and shaded area shows kernel density estimation for the data.

cDNA in the intestine (**Figure 5B**). Thus, TORC2 signaling in the intestine is a key repressor of *str-44* expression in AWA. Consequently, *rict-1* may be part of a pathway that detects internal nutritional state information and modulates AWA chemoreceptor expression.

To identify additional components of the TORC2 signaling pathway that regulates *str-44*, we performed a feeding RNAi screen against known members of the TORC2 pathway (*akt-1*, *let-363*, *pkc-2*, *sgk-1*, *sinh-1*, and *skn-1*). Of these, *sgk-1(RNAi)* and *pkc-2(RNAi)* produced an elevation of *str-44* expression similar to *rict-1(RNAi)* (**Figure 5C**). The TORC2 complex is known to phosphorylate and activate *sgk-1* and *pkc-2* (**Jones et al., 2009**), suggesting that *rict-1* likely acts through these effectors to suppress *str-44* expression in AWA. We also examined whether loss of any of these genes could suppress the elevated *str-44* expression in *rict-1* mutants and found that *skn-1(RNAi)* reduced *str-44* expression in this background (**Figure 5C**). This observation is consistent with previous findings showing that *skn-1* encodes a transcription factor, homologous to Nrf1/2, that is inhibited by *rict-1* signaling (**Ruf et al., 2013**). Together, these experiments reveal that the nutrient sensing TORC2 pathway functions in the intestine to regulate olfactory receptor expression in AWA.

## The pathways that converge upon AWA control olfactory receptor expression and behavior in a modular manner

The experiments above identify molecular and neural pathways that allow external and internal cues to influence olfactory receptor expression in AWA. To understand how AWA integrates these signals, we next examined how they interact. We first investigated the pathways that were implicated in crosstalk from food sensory neurons to AWA. Given that the food sensory neurons that inhibit *str-44* expression release insulin peptides, and that the insulin pathway in AWA also influences *str-44* expression, we tested whether the sensory neurons signal to AWA via DAF-2/DAF-16 signaling. Indeed, the elevated expression of *str-44* in *tax-4* sensory-defective mutants or in animals with the ASI sensory neuron silenced was suppressed by the loss of the *daf-16*/FOXO transcription factor, a key target of the insulin pathway and other signaling pathways (**Figure 6A**). This suggests that elevated *str-44* expression due to inactivation of food sensory neurons requires downstream *daf-16*/FOXO signaling. We also found that the enhanced expression of *str-44* caused by hyperactive *goa-1* signaling in AWA was fully suppressed by a *daf-16* mutation (**Figure 6A**). In addition, silencing of AWA, which reduces *str-44* expression in wild-type animals, had no effect in a *daf-2*/InR mutant (**Figure 6—figure supplement 1**). Together, these experiments indicate that food sensory neurons signal to AWA via DAF-2-DAF-16 insulin signaling. Changes in AWA activity and G protein signaling that modulate *str-44* expression also depend on the DAF-2-DAF-16 pathway.

We next examined whether osmotic stress, which decreases *str-44* expression, operates through any of the pathways identified. Thus, we examined whether osmotic stress during fasting could suppress the elevated *str-44* expression seen in mutant animals lacking *tax-4*, *daf-2*, or *rict-1*. Osmotic stress still had an effect in all of these backgrounds (**Figure 6B**), suggesting that osmotic stress inhibits *str-44* expression through an as yet unidentified pathway that does not require *tax-4, daf-2,* or *rict-1*.

Next, we examined how the intestinal *rict-1*/Rictor pathway interacts with the sensory neuron-insulin signaling pathway. A double mutant lacking both *rict-1*/Rictor and *daf-2*/InR showed significantly greater levels of *str-44* expression than each of the single mutants (**Figure 6C**), suggesting that these two pathways function in parallel to inhibit *str-44* expression. Interestingly, we found that a *rict-1;daf-16* double mutant displayed a phenotype matching *daf-16* single mutants (**Figure 6C**). This suggests that modulation of *str-44* expression by the intestinal TORC2 pathway requires downstream *daf-16*/FOXO signaling in AWA. Given that *rict-1* and *daf-2* act in parallel (the *rict-1;daf-2* double mutant phenotype is more severe than that of the single mutants), *rict-1* likely modulates *daf-16* function through a noninsulin pathway. Together, these results identify several parallel pathways that converge on AWA to control *str-44* expression. These pathways appear to act in a modular manner where they can each independently influence chemoreceptor expression (illustrated in **Figure 6D**). Regulation of *str-44* by several of these stimuli depends on *daf-16*/FOXO, suggesting that it might serve as a molecular locus of integration.

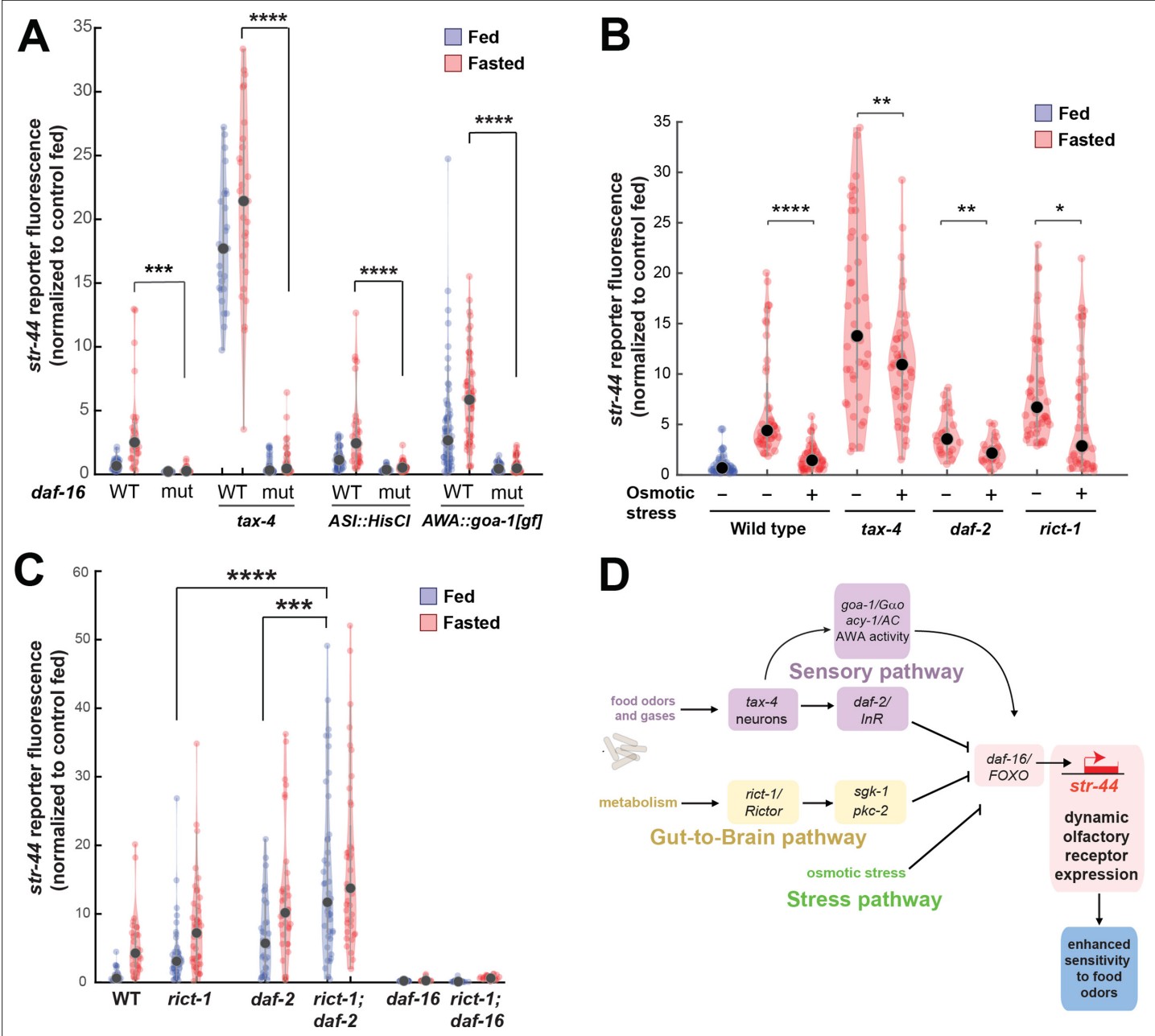

**Figure 6.** Food sensory signals, gut-to-brain metabolic signals, and stress pathways act in a parallel, modular fashion to control *str-44* expression. (**A**) Relative expression of *str-44*::T2A-mNeonGreen in wild-type, *daf-16*, *tax-4*, *tax-4;daf-16*, ASI::HisCl. ASI::HisCl;daf-16, AWA::goa-1, AWA::goa-1;daf-16 animals. (**B**) Relative expression of *str-44*::T2A-mNeonGreen in wild-type, *tax-4*, *daf-2*, and *rict-1* mutant animals fasted on standard nematode growth medium (NGM) plates (150 mOsm) or high osmolarity NGM plates (300 mOsm). (**C**) Relative expression of str-44::T2A-mNeonGreen in *rict-1*, *daf-2*, *rict-1;daf-2*, *daf-16*, and *rict-1;daf-16* mutants compared to wild-type controls. (**D**) Schematic depicting the parallel pathways that converge on AWA to modulate *str-44* expression, which in turn modulates food-seeking behavior. For (**A–C**), each condition measured in two independent sessions. ****p<0.0001, ***p<0.001, **p<0.01, *p<0.05 by two-tailed *t*-test with Bonferroni correction. Colored dots represent from individual cells, black dots represent median values, and shaded area shows kernel density estimation for the data.

The online version of this article includes the following figure supplement(s) for figure 6:

**Figure supplement 1.** Relative expression of *str-44*::T2A-mNeonGreen in *daf-2*, AWA::HisCl, and AWA::HisCl;daf-2 mutants compared to wild-type control.

## Discussion

Animals respond to sensory cues by generating behavioral responses that reflect their ongoing needs and states. Yet how sensorimotor circuits integrate diverse cues relevant to these needs and modulate their function accordingly is poorly understood. We find that a single *C. elegans* olfactory neuron integrates multiple states and stimuli to influence its expression of olfactory receptors, which in turn alters the animal's food-seeking behaviors. Several molecular and neural pathways that originate in different cell types throughout the body converge on AWA to regulate olfactory receptor expression: crosstalk from other sensory neurons, metabolic signals from the gut, and pathways that signal physiological stress. Our behavioral findings show that the synthesis of these signals by AWA allows animals to generate sensorimotor responses that reflect the animal's overall state. These results suggest a general model in which sensory- and state-dependent transcriptional changes at the sensory periphery modulate animals' sensorimotor responses to meet their ongoing needs and states.

We found that AWA olfactory receptor expression reflects recent sensory stimuli, metabolic state, and physiological stress. Recent sensory stimuli are detected by a set of food sensory neurons – ASI, ASK, AWB, BAG, or AQR/PQR/URX – that inhibit AWA olfactory receptor expression in the presence of food. Our data suggest that one possible route of signaling involves the food sensory neurons synapsing onto a second-order neuron in the circuit, AIA, which in turn synapses onto AWA. In addition, sensory neurons may release insulin-like peptides that can activate the DAF-2 insulin receptor in AWA. Given that our experiments suggest a surprisingly high level of crosstalk among the sensory neurons, one possibility is that AWA chemoreceptor expression may reflect the integration of activity across the full chemosensory circuit.

AWA also integrates metabolic signals from the gut. The TORC2 complex that responds to nutrient levels in the intestine (*O'Donnell et al., 2018*; *Soukas et al., 2009*) appears to regulate an as yet unidentified gut-to-brain signaling pathway that impacts AWA olfactory receptor expression. The TORC2 pathway has been shown to impact *daf-28*/insulin-like peptide expression (*O'Donnell et al., 2018*). However, the *rict-1* mutation enhances *str-44* expression in the *daf-2*/InR mutant, suggesting that *rict-1* also operates through an unidentified pathway that is independent of the insulin pathway.

Our data are most consistent with a framework in which diverse signaling pathways converge on AWA chemoreceptor expression to allow animals to generate sensorimotor behaviors that may vary across environmental conditions. The modulation of chemoreceptor expression in AWA is unlikely to be explained by a simple homeostatic mechanism in which the purpose of altering chemoreceptors would be to keep AWA activity at a target set point. If this mechanism were in effect, then direct inhibition of AWA activity would be expected to increase the expression of excitatory chemoreceptors; however, we found the opposite to be the case (*Figure 4E*). Sensory habituation alone is unlikely to be an explanation for the data as exposure to *str-44* odorants does not modify *str-44* expression (*Figure 4—figure supplement 2B*). In addition, our results cannot be explained by a model where *str-44* expression simply tracks AWA activity as we find that the high expression level of *str-44* in fasted *daf-2* mutants was not reduced by inhibiting AWA activity (*Figure 6—figure supplement 1*).

Although this study focuses largely on two state-dependent chemoreceptors that are expressed in one sensory neuron, our results suggest that >100 olfactory receptors undergo similar state-dependent regulation across multiple classes of sensory neurons (*Figure 1*). In addition, previous work has shown that the GPCRs *odr-10* and *srh-234* undergo feeding state-dependent regulation (*Gruner et al., 2014*; *Wexler et al., 2020*). As *C. elegans* have a relatively small number of olfactory neurons, receptors are an attractive site for sensory flexibility. Consistent with this notion, chemoreceptor genes are frequent targets of evolution that drive naturally occurring changes in behavior (*Baldwin et al., 2014*; *Greene et al., 2016*; *Nei et al., 2008*). Multiple odorants typically activate a single olfactory receptor. Thus, changing the expression of only a few genes could alter responses to many odors. In their natural environment, *C. elegans* are found in soil, compost, or ripe/rotting fruits (*Frézal and Félix, 2015*). Interestingly, the odors that we identified as ligands for STR-44, propyl and butyl acetate, are major components of the aroma of ripe fruits (*López et al., 1998*). We also found that *str-44* and *srd-28* drive AWA responses to bacterial odor mixtures. Fruit odors and bacterial odors can both suggest the presence of nearby food to *C. elegans*. The broad tuning of STR-44 to these food and food-adjacent stimuli may allow fasted animals to maximize their chances of encountering food. We expect that other chemoreceptors will be modulated in a similar fashion to control sensory neuron responses and stimulus-specific foraging behaviors.

In addition, we found that a large number of non-chemoreceptor neuronal genes are differentially expressed in response to fasting (*Supplementary file 1*). This result complements previous findings that changes in gene expression are widespread following fasting and in mutants lacking fasting-responsive transcription factors (*Harvald et al., 2017*; *Kaletsky et al., 2016*). It is likely that other genes in our dataset contribute to feeding state-dependent changes in neuronal activity and behavior.

Our results suggest a potential mechanism by which animals can generate behaviors that reflect an integration of multiple ongoing needs and states. Individual states, like hunger or mating drive, are represented by devoted cell populations that impact many aspects of behavior. For animals to generate behaviors that reflect their overall state, signals from diverse sources need to be integrated by the sensorimotor circuits that implement motivated behaviors. We find that convergent signaling onto neurons in sensorimotor circuits modulates gene expression and thus alters circuit function over long timescales. The inputs onto AWA that convey state information arise from a variety of cell types throughout the body but converge on the *daf-16*/FOXO transcription factor in AWA that controls gene expression. We found that these parallel pathways act in a modular fashion where they can each independently influence chemoreceptor expression, which likely allows animals to adaptively tune their expression of chemoreceptors depending on the sensory cues, stressors, and nutrients in the environment. It is likely that many other neurons in the sensorimotor circuits of *C. elegans* and other animals similarly integrate a wide range of state-relevant inputs to modify their gene expression programs and functional properties.

# Materials and methods

## Key resources table

| Reagent type (species) or resource | Designation | Source or reference | Identifiers | Additional information |
|---|---|---|---|---|
| Strain, strain background (*Caenorhabditis elegans*) | Wild-type Bristol N2 | N/A | N2 | |
| Genetic reagent (*C. elegans*) | odr-7(ky4) | *Sengupta et al., 1994* | CX4 | |
| Genetic reagent (*C. elegans*) | odr-7(ky4); tax-4(p678) | This study | SWF482 | Strain available from Flavell Lab (see 'Data availability') |
| Genetic reagent (*C. elegans*) | kyIs665[rimb-1::rpl-22-3xHA,myo-3::mCherry] | This study | CX16283 | Strain available from Flavell Lab (see 'Data availability') |
| Genetic reagent (*C. elegans*) | kyIs587[gpa-6::GCaMP2.2b, unc-122::dsRed] | *Larsch et al., 2013* | NZ1101 | |
| Genetic reagent (*C. elegans*) | srd-28(syb2320) | This study | PHX2320 | Strain available from Flavell Lab (see 'Data availability') |
| Genetic reagent (*C. elegans*) | str-44(syb1869) | This study | PHX1869 | Strain available from Flavell Lab (see 'Data availability') |
| Genetic reagent (*C. elegans*) | odr-10(syb3508) | This study | PHX3508 | Strain available from Flavell Lab (see 'Data availability') |
| Genetic reagent (*C. elegans*) | ser-7(syb1941) | This study | PHX1941 | Strain available from Flavell Lab (see 'Data availability') |
| Genetic reagent (*C. elegans*) | str-44(syb1869); cat-1(e1111) | This study | SWF428 | Strain available from Flavell Lab (see 'Data availability') |
| Genetic reagent (*C. elegans*) | str-44(syb3563)IV; srd-28(syb3336)V | This study | PHX3677 | Strain available from Flavell Lab (see 'Data availability') |
| Genetic reagent (*C. elegans*) | str-44(syb1869);daf-3(e1376) | This study | SWF456 | Strain available from Flavell Lab (see 'Data availability') |
| Genetic reagent (*C. elegans*) | str-44(syb1869);pdfr-1(ok3425) | This study | SWF461 | Strain available from Flavell Lab (see 'Data availability') |
| Genetic reagent (*C. elegans*) | odr-7(ky4); tax-4(p678); flvEx283[sra-6::srd-28 cDNA, myo2::mCherry] | This study | SWF631 | Strain available from Flavell Lab (see 'Data availability') |
| Genetic reagent (*C. elegans*) | odr-7(ky4); tax-4(p678); flvEx181[sra-6::str-44 cDNA, myo-3::mCherry] | This study | SWF478 | Strain available from Flavell Lab (see 'Data availability') |

*Continued on next page*

*Continued*

| Reagent type (species) or resource | Designation | Source or reference | Identifiers | Additional information |
|---|---|---|---|---|
| Genetic reagent (*C. elegans*) | tax-4(p678); str-44(syb1869) | This study | SWF486 | Strain available from Flavell Lab (see 'Data availability') |
| Genetic reagent (*C. elegans*) | str-44(syb1869); ceh-36(ks86) | This study | SWF514 | Strain available from Flavell Lab (see 'Data availability') |
| Genetic reagent (*C. elegans*) | daf-7(e1372); str-44(syb1869) | This study | SWF522 | Strain available from Flavell Lab (see 'Data availability') |
| Genetic reagent (*C. elegans*) | daf-2(m41); str-44(syb1869) | This study | SWF527 | Strain available from Flavell Lab (see 'Data availability') |
| Genetic reagent (*C. elegans*) | str-44(syb1869); flvEx216[Pgpa-6::HisCl] | This study | SWF528 | Strain available from Flavell Lab (see 'Data availability') |
| Genetic reagent (*C. elegans*) | str-44(syb1869); flvEx239[srh-11::HisCl1-sl2-mCherry(25 ng/µL)] | This study | SWF557 | Strain available from Flavell Lab (see 'Data availability') |
| Genetic reagent (*C. elegans*) | str-44(syb1869); raga-1(ok386) | This study | SWF545 | Strain available from Flavell Lab (see 'Data availability') |
| Genetic reagent (*C. elegans*) | str-44(syb1869); rict-1(ft7) | This study | SWF546 | Strain available from Flavell Lab (see 'Data availability') |
| Genetic reagent (*C. elegans*) | str-44(syb1869); flvEx231[srg-47p::HisCl1-sl2-mCherry+myo-3p::mCherry] | This study | SWF548 | Strain available from Flavell Lab (see 'Data availability') |
| Genetic reagent (*C. elegans*) | str-44(syb1869); flvEx238[tax-4::tax-4 (40 ng/µL)+myo-2::mCherry (1 ng/µL)] | This study | SWF552 | Strain available from Flavell Lab (see 'Data availability') |
| Genetic reagent (*C. elegans*) | str-44(syb1869); flvEx239[srh-11::HisCl1-sl2-mCherry(25 ng/µL)] | This study | SWF557 | Strain available from Flavell Lab (see 'Data availability') |
| Genetic reagent (*C. elegans*) | str-44(syb1869); flvEx235 [gcy-33::HisCl1-sl2-mCherry (7.5 ng/µL)] | This study | SWF556 | Strain available from Flavell Lab (see 'Data availability') |
| Genetic reagent (*C. elegans*) | str-44(syb1869); flvEx245[gcy-36::HiCl-sl2-mCherry (2 ng/µL)] | This study | SWF563 | Strain available from Flavell Lab (see 'Data availability') |
| Genetic reagent (*C. elegans*) | str-44(syb1869); flvEx246[str-1::HiCl-sl2-mCherry (25 ng/µL)] | This study | SWF564 | Strain available from Flavell Lab (see 'Data availability') |
| Genetic reagent (*C. elegans*) | str-44(syb1869); flvEx241[sra-9::HiCl-sl2-mCherry (25 ng/µL)] | This study | SWF559 | Strain available from Flavell Lab (see 'Data availability') |
| Genetic reagent (*C. elegans*) | daf-2(m41); str-44(syb1869); flvEx258[gpa-6::daf-2-sl2-mCherry (25 ng/µL)] | This study | SWF583 | Strain available from Flavell Lab (see 'Data availability') |
| Genetic reagent (*C. elegans*) | daf-2(m41); str-44(syb1869); flvEx263[gpa-6::daf-2-sl2-mCherry (25 ng/µL)] | This study | SWF588 | Strain available from Flavell Lab (see 'Data availability') |
| Genetic reagent (*C. elegans*) | str-44(syb1869); mxl-3(ok1947) | This study | SWF589 | Strain available from Flavell Lab (see 'Data availability') |
| Genetic reagent (*C. elegans*) | str-44(syb1869); flvEx272[gpa-6::acy-1(gf) (25 ng/µL), myo-2::mCherry (1 ng/µL)] | This study | SWF611 | Strain available from Flavell Lab (see 'Data availability') |
| Genetic reagent (*C. elegans*) | str-44(syb1869); flvEx279[25 ng/µL gpa-6::goa-1(GF)-sl2-mCherry] | This study | SWF622 | Strain available from Flavell Lab (see 'Data availability') |
| Genetic reagent (*C. elegans*) | str-44(syb1869); daf-16(mu86); flvEx278[25 ng/µL gpa-6::daf-16a-sl2-mCherry] | This study | SWF621 | Strain available from Flavell Lab (see 'Data availability') |
| Genetic reagent (*C. elegans*) | str-44(syb1869); daf-16(mu86); flvEx278[25 ng/µL dpy-30::daf-16a-sl2-mCherry] | This study | SWF625 | Strain available from Flavell Lab (see 'Data availability') |
| Genetic reagent (*C. elegans*) | str-44(syb1869); daf-16(mu86) | This study | SWF627 | Strain available from Flavell Lab (see 'Data availability') |
| Genetic reagent (*C. elegans*) | flvEx284[ges-1::rict-1+myo-3::mCherry]; str-44(syb1869); rict-1(ft7) | This study | SWF633 | Strain available from Flavell Lab (see 'Data availability') |
| Genetic reagent (*C. elegans*) | daf-16(mu86); tax-4(p678); str-44(syb1869) | This study | SWF638 | Strain available from Flavell Lab (see 'Data availability') |

*Continued on next page*

*Continued*

| Reagent type (species) or resource | Designation | Source or reference | Identifiers | Additional information |
|---|---|---|---|---|
| Genetic reagent (*C. elegans*) | *daf-16(mu86); rict-1(ft7); str-44(syb1869);* | This study | SWF642 | Strain available from Flavell Lab (see 'Data availability') |
| Genetic reagent (*C. elegans*) | *str-44(syb1869); flvEx292[ceh-36p::TeTx-SL2-mCherry+myo-3p::mCherry]* | This study | SWF648 | Strain available from Flavell Lab (see 'Data availability') |
| Genetic reagent (*C. elegans*) | *str-44(syb1869); flvEx293[gpa-6p::egl-30(gf)+myo-3p::mCherry]* | This study | SWF649 | Strain available from Flavell Lab (see 'Data availability') |
| Genetic reagent (*C. elegans*) | *daf-16(mu86); str-44(syb1869); flvEx231[srg-47p::HisCl1-sl2-mCherry+myo-3p::mCherry]* | This study | SWF666 | Strain available from Flavell Lab (see 'Data availability') |
| Genetic reagent (*C. elegans*) | *str-44(syb1869); daf-2(m41); flvEx216[gpa-6::HisCl1]* | This study | SWF668 | Strain available from Flavell Lab (see 'Data availability') |
| Genetic reagent (*C. elegans*) | *str-44(syb1869); tax-4(p678); flvEx216[gpa-6::HisCl1]* | This study | SWF669 | Strain available from Flavell Lab (see 'Data availability') |
| Genetic reagent (*C. elegans*) | *str-44(syb1869); rict-1(ft7); flvEx216[gpa-6::HisCl1]* | This study | SWF670 | Strain available from Flavell Lab (see 'Data availability') |
| Genetic reagent (*C. elegans*) | *str-44(syb1869); aak-1(tm1944); aak-2(gt33)* | This study | SWF671 | Strain available from Flavell Lab (see 'Data availability') |
| Genetic reagent (*C. elegans*) | *str-44(syb1869);daf-16(mu86); flvEx297[gpa-6::goa-1(gf)-sl2-mCherry (25 ng/µL)]* | This study | SWF672 | Strain available from Flavell Lab (see 'Data availability') |
| Genetic reagent (*C. elegans*) | *str-44(syb1869);daf-7(e1372); flvEx297[gpa-6::goa-1(gf)-sl2-mCherry (25 ng/µL)]* | This study | SWF678 | Strain available from Flavell Lab (see 'Data availability') |
| Genetic reagent (*C. elegans*) | *str-44(syb1869); rict-1(ft7); daf-7(e1372)* | This study | SWF679 | Strain available from Flavell Lab (see 'Data availability') |
| Genetic reagent (*C. elegans*) | *str44(syb1869); atgl-1(gk176565)* | This study | SWF637 | Strain available from Flavell Lab (see 'Data availability') |
| Genetic reagent (*C. elegans*) | *str44(syb1869); lim-4(ky403)* | This study | SWF687 | Strain available from Flavell Lab (see 'Data availability') |
| Genetic reagent (*C. elegans*) | *str-44(syb1869); rict-1(ft7); daf-2 (m41)* | This study | SWF699 | Strain available from Flavell Lab (see 'Data availability') |
| Genetic reagent (*C. elegans*) | *str-44(syb1869); flvEx350[gpa-6::egl-30(gf) (5 ng/µL), myo-2::mCherry (1 ng/µL)]* | This study | SWF750 | Strain available from Flavell Lab (see 'Data availability') |
| Genetic reagent (*C. elegans*) | *str-44(syb1869); rict-1(ft7); daf-16(mu86)* | This study | SWF642 | Strain available from Flavell Lab (see 'Data availability') |
| Genetic reagent (*C. elegans*) | *flvEx390[gpa-6::str-44-sl2-mCherry]* | This study | SWF827 | Strain available from Flavell Lab (see 'Data availability') |
| Genetic reagent (*C. elegans*) | *flvEx308[ttx-3::TeTx, myo-3::mCherry]* | This study | SWF698 | Strain available from Flavell Lab (see 'Data availability') |
| Genetic reagent (*C. elegans*) | *flvEx307[gcy-28d::TeTx, myo-3::mCherry]* | This study | SWF697 | Strain available from Flavell Lab (see 'Data availability') |
| Genetic reagent (*C. elegans*) | *flvEx399[gpa-6::TeTx, myo-3::mCherry]* | This study | SWF843 | Strain available from Flavell Lab (see 'Data availability') |
| Genetic reagent (*C. elegans*) | *srd-28(syb2320); daf-16(mu86)* | This study | SWF844 | Strain available from Flavell Lab (see 'Data availability') |
| Genetic reagent (*C. elegans*) | *srd-28(syb2320); daf-2(m41)* | This study | SWF845 | Strain available from Flavell Lab (see 'Data availability') |
| Genetic reagent (*C. elegans*) | *srd-28(syb2320); tax-4(p678)* | This study | SWF846 | Strain available from Flavell Lab (see 'Data availability') |
| Chemical compound, drug | Butyl acetate | Sigma-Aldrich | 287725 | |
| Chemical compound, drug | Propyl acetate | Sigma-Aldrich | 537438 | |
| Chemical compound, drug | Aztreonam | Sigma-Aldrich | PZ0038 | |
| Software, algorithm | MATLAB | MathWorks | R2014a, R2021b | |

## Growth conditions and handling

Nematode culture was conducted using standard methods. Populations were maintained on nematode growth medium (NGM) agar plates with *E. coli* OP50 bacteria. Wild-type was *C. elegans* Bristol strain N2. For genetic crosses, genotypes were confirmed using PCR. Transgenic animals were generated by injecting DNA clones plus fluorescent co-injection marker into gonads of young adult hermaphrodites. One-day-old hermaphrodites were used for all assays. All assays were conducted at room temperature (~22°C). Plates for the osmotic stress experiments were normal NGM plus 150 mM sorbitol added. In the presence of this mild stressor, animals displayed altered behaviors such as egg-laying, but do not display acute reversal responses and grow at normal rates (*Yu et al., 2017*; *Zhang et al., 2008*).

## Plasmid construction

For HisCl1-based silencing of neurons, we inserted the following promoters into pSM-HisCl1-sl2-mCherry: *gpa-6* (AWA), *str-1* (AWB), *gcy-33* (BAG), *srg-47* (ASI), *gcy-36* (URX/AQR/PQR), *sra-9* (ASK), *srh-11* (ASJ). For TeTx-based silencing, we inserted the following promoters into pSM-TeTx-sl2-mCherry: *ceh-36*short (AWC), *gcy-28.d* (AIA), *ttx-3* (AIY).

For AWA-specific expression of HisCl1, *acy-1(gf)*, *goa-1(gf)*, *egl-30(gf)*, *srd-28*, *str-44*, *daf-2*, and *daf-16,* we used the *gpa-6* promoter. *goa-1(gf)* and *egl-30(gf)* were synthesized with the Q205L gain-of-function mutations added at the time of synthesis. These cDNAs were subsequently subcloned into pSM. The *acy-1(gf)* clone was described in a previous study (*Flavell et al., 2013*). *srd-28* and *str-44* were amplified from pooled cDNA and inserted into the pSM vector. For ASH-specific expression of *str-44* and *srd-28*, we used the *sra-6* promoter. The *daf-16* rescue plasmid was generated by PCR amplifying the *daf-16.a* cDNA from pooled cDNA and inserting it into pSM. The *daf-2* rescue plasmid was generated by excising the *daf-2* cDNA from pJH4531 (a kind gift from M. Zhen) and inserting it into pSM.

The *tax-4* and *rict-1* rescue plasmids have been previously described (*Macosko et al., 2009*; *O'Donnell et al., 2018*).

## CRIPSR gene editing

The T2A-mNeonGreen reporter strains were constructed by inserting a T2A-mNeonGreen coding sequence immediately before the stop codons of *str-44, srd-28, odr-10,* and *ser-7* via CRISPR/Cas9 gene editing.

The *str-44* mutant was created by introducing an indel at the start codon of the *str-44* gene, resulting in deletion of the start codon. The next internal methionine in *str-44* occurs in the second transmembrane domain and there are no upstream sequences that could result in an in-frame start codon, suggesting that this should result in a null mutation. The *srd-28* mutant was created by introducing a frameshift indel near the beginning of the second coding exon of *srd-28*, which is a large exon that encodes three of the seven transmembrane domains.

## Translating ribosome affinity purification and analysis

Translating ribosome affinity purification was performed as described in a previously published detailed protocol (*McLachlan and Flavell, 2019*). Briefly, a ribotagging plasmid was constructed containing the *C. elegans rpl-22* cDNA with three tandem HA tags under control of the *rimb-1* (previously *tag-168*) pan-neuronal promoter. Animals containing an integrated copy of this transgene were grown on 15 cm enriched-peptone (20 g/L) NGM plates seeded throughout with OP50 to 1-day old adults, then washed to fresh plates with or without OP50 lawns seeded 1 day prior. After 3 hr, animals were collected with liquid NGM supplemented with cycloheximide (0.8 mg/mL) and flash frozen within minutes. Samples were then prepared for lysis and RNA isolation as previously described. We performed three independent biological replicates in total.

Whole animal and ribotag RNA samples were amplified with the Clontech SMART-Seq v2 Low Input RNA kit and prepared as Illumina Nextera XT libraries by the MIT BioMicroCenter sequencing core. Reads were mapped to the *C. elegans* genome (WBcel235) with kallisto (*Bray et al., 2016*) and analyzed for differential expression with sleuth (*Pimentel et al., 2017*) and custom scripts. Data are deposited at GEO accession number GSE200640. For the data shown in *Figure 1* and *Supplementary file 1*, we required that each enriched gene was fourfold enriched in the differential expression analysis.

## Confocal imaging and quantification

For experiments using the *str-44*p::mNeonGreen reporter, animals were imaged with the same laser power, exposure time, and objective lens to allow for comparisons between experimental conditions. For each experimental condition, 20–30 animals were immobilized in 5 mM tetramisole hydrochloride (Sigma) on a #1.5 coverslip, then mounted on slides with minimally thick NGM pads. Data were collected on a Nikon Eclipse Ti microscope coupled to a Yokogawa CSU-X1 spinning disk unit with a Borealis upgrade. We used a ×40/1.15NA CFI Apo LWD Lambda water immersion objective and NIS Elements software for data acquisition. Z-stacks were collected through the entire depth of the animal at 0.5 micron steps. Fluorescence intensity was quantified in ImageJ (NIH). For each animal, a maximum intensity z-projection containing the neuron nearest the objective was generated, a box was drawn around the neuron, and an intensity profile was generated from the box. A single background-subtracted fluorescence intensity value was calculated for each cell by subtracting the mean bottom 5% of fluorescent signal (background) from the mean top 5% of fluorescent signal (neural signal). To aid comparisons between experiments, these values were normalized to the mean of fed wild-type control animals imaged on the same day. Violin plots were generated with a custom MATLAB function (Bastian Bechtold, Violin Plots for MATLAB, https://github.com/bastibe/Violinplot-Matlab; *García, 2022*).

## Food choice assay

The binary food choice assay was performed as previous described (*Worthy et al., 2018*) with minor modifications. OP50 bacteria were grown overnight with agitation in LB media at 37°C, centrifuged at 5000 rpm for 2 min, and resuspended in fresh LB to OD600 = 10. Then 6 cm NGM plates were spotted with two drops of 25 µL each, air dried for 10 min, and covered with a lid and incubated at room temperature (22°C) for 5 hr. Also, 2 µL of odorant or ethanol vehicle was added adjacent to the bacterial lawns immediately prior to adding animals to the plate. Adult animals were washed twice in S. Basal buffer and 40–200 animals were placed near the center of the plate, equidistant from the bacterial patches, and the plate was covered with a lid. Following standard protocols (*Worthy et al., 2018*), animals were immobilized by adding 5 µL of 1 M sodium azide to each bacterial patch after 1 hr. Animals inside each patch were counted, and a food choice index was calculated as animals within experimental patch/nimals within both patches.

## Exploration assays and recordings of locomotion

Behavioral assays for on-food exploration were conducted as previously described (*Flavell et al., 2013*) with minor modifications. One-day-old adults were washed off growth plates and fed on OP50 lawns (seeded the day before) or fasted off-food for 3 hr. Individual animals were then picked to 60 mm plates uniformly seeded with OP50 and allowed to freely locomote for 5 hr. After this time, animals were removed from the lawn, and plates were superimposed on a grid containing 3.5 mm squares and the number of squares containing worm tracks was manually counted.

For quantification of animal speed in the absence of food, 1-day-old adult animals were washed from OP50 plates with liquid NGM and, after three washes, were transferred to 10 cm NGM plates with copper filter paper (Whatman paper soaked in 0.02 M CuCl$_2$) boundaries. Animals were recorded for 1 hr on JAI Spark SP-20000M cameras with Streampix 7 software at 3 fps, and speed was extracted from videos using custom MATLAB scripts as previously described (*Rhoades et al., 2019*).

## Culture and use of diverse bacterial species

For experiments involving species other than *E. coli* OP50, all bacterial species (PA14, DA1877, JUB19, MYB71, BigB0393) were streaked onto LB plates from frozen stocks and plates were incubated overnight at 25°C. A sterile pipette tip was used to pick colonies from the plate into LB medium for an overnight incubation at 37°C. NGM plates were seeded with 200 µL of the bacteria respectively for the feeding experiments followed by confocal imaging as described above. JUB19, MYB71, and BigB0393 are courtesy of the CeMbio collection at the Caenorhabditis Genetics Center.

For food on lid (odor) experiments done on PA14 and DA1877, bacteria were cultured as described above. A thin NGM pad was placed on the lid with 150 µL of OD600 = 1 bacteria. This was chosen due to previous work on PA14 inducing aversive olfactory learning that used this same concentration (*Zhang et al., 2005*).

## Feeding with aztreonam-treated bacteria

To aztreonam-treat bacteria, a standard OP50 culture was diluted fivefold in LB with 5 µg/mL aztreonam and grown overnight at 37°C with gentle agitation. Bacteria were then plated on NGM plates with 10 µg/mL aztreonam. Bacteria were allowed to grow overnight at room temperature, then visually inspected for filamentous growth before animals were plated for the experiment.

## RNA interference screen

Animals were fed on bacteria containing RNAi feeding vectors from the Ahringer (*Kamath et al., 2003*) (for *unc-22, akt-1, skn-1, rict-1,* and *let-363*) or ORFeome (*Rual et al., 2004*) (for *sinh*-1, *sgk*-1, and *pkc-2*) libraries. Bacteria were plated in a uniform lawn on 6 cm NGM plates with the addition of 25 µg/mL carbenicillin and 1 mM IPTG. To avoid maternal effect, embryonic lethality, and early developmental defects from RNAi, as well as starvation associated with standard L1 synchronization, *str-44*::T2A-mNeonGreen animals were allowed to lay eggs on standard NGM plates seeded with OP50, then the hatched progeny were washed with M9 onto feeding RNAi plates at L1-L2 stage. Animals were imaged as young adults (~2 days later) only when *unc-22* positive control plates produced twitching animals.

## Chemotaxis assays

Chemotaxis assays were performed as previously described (*Bargmann et al., 1993*). Assays were conducted on square grid plates of assay agar, poured the night before the assay. Assays were conducted in a 22°C incubator set to 40% humidity. One-day-old adult animals were washed off growth plates using S. Basal buffer, then washed twice more with S. Basal and once with water. A total of 50–200 animals were placed on the chemotaxis plates. Two 1 µL spots of odor were placed on one side of the plate, and two 1 µL spots of ethanol on the opposite side (ethanol was used as the dilutant for all odors). Two 1 µL spots of 1 M sodium azide were placed on either end of the plate to paralyze animals at the odor source. Animals navigated the plate for 60 min (except 90 min for the *tax-4; odr-7* worms and those in that background, as their locomotion was slower). Plates were then placed in a 4°C cold room to arrest movement. The assay was scored by counting animals that arrived at the odor, at the control ethanol spot, or elsewhere in the plate. This was used to calculate a Chemotaxis Index $(\#_{odor} - \#_{ethanol})/(\#_{odor} + \#_{ethanol} + \#_{other})$. Worms in the center of the assay plate that did not move from their starting position were excluded.

Odor concentrations used were as follows: 1:1000 diacetyl, 1:1000 methyl pyrazine, 10 mg/mL pyrazine, 1:10 butyl acetate, 1:10 propyl acetate, 1:100 hexyl acetate, 1:100 ethyl acetate, and 1:100 isoamyl acetate.

## Calcium imaging in freely moving animals

In vivo calcium imaging of AWA::GCaMP2.2b in freely moving animals was carried out on a Nikon Eclipse Ti-S microscope with a ×2/0.10NA Plan Apo objective and an Andor Zyla 4.2 Plus sCMOS camera. Blue light application to animals from an X-Cite 120LED system was 10 ms for each exposure, at a frame rate of 10 fps. For these experiments, slides were prepared by placing a 2 µL drop of OP50 (OD600 = 2) on a minimally thick NGM pad, then a custom cut PDMS corral was placed on the pad. Five animals were picked (without food) to the center of the corral (off food) and covered with a coverslip, then immediately imaged. Animals began outside the field of view of the objective lens and were allowed to freely navigate into the bacterial lawn within the field of view. Individual slides were imaged for no longer than 30 min. Background-subtracted intensity values for AWA were extracted from each video frame using custom ImageJ macros, as described previously (*Flavell et al., 2013*). Food patch encounter frames were manually annotated. $\Delta F/F_0$ was calculated as (fluorescence – baseline)/baseline, where fluorescence is a 10 frame (1 s) moving median and baseline is the 5th percentile fluorescence throughout the recording.

## Calcium imaging during odor delivery in immobilized animals

Imaging of neurons was performed as previously described (*Chute et al., 2019*). AWA::GCaMP2.2b animals were picked as larval stage 4 (L4) hermaphrodites the day prior to imaging and singled onto an NGM plate seeded with OP50. The animals were kept at 20°C for 16 hr. The next day they were imaged as young adults. Animals were loaded into a modified PDMS olfactory chip (*Reilly et al.,*

*2017*) only allowing the animal's nose to be subjected to the solution. Well-fed worms were transferred directly from the NGM plate into the chip. Fasted animals were transferred to an unseeded NGM plate 3 hr prior to imaging and then transferred into the chip. Animals were imaged under ×40 magnification for two 30 s trials. Each trial consisted of a 5 s period prior to stimulation, a 10 s odor pulse, and a 15 s recording post stimulation. The recording was performed in Micro-Manager recording TIFF stacks at 10 frames/s, exciting the neuron with blue light at 470 nm. A minimum of 10 animals (or 20 trials) was captured for each condition.

The solutions used for imaging were all made in S. Basal (100.103 mM NaCl, 5.741 mM $K_2HPO_4$, 44.090 mM $KH_2PO_4$, 0.0129 mM cholesterol in $H_2O$). The solvent control solution was 1 mM tetramisole, 0.3 µM fluorescein in S. Basal. The flow control solution, not exposed to the worm but controlling the movement of the solutions in the olfactory chip, was 1 mM tetramisole, 0.6 µM fluorescein in S. Basal. The stimuli were $10^{-6}$ propyl acetate or butyl acetate, prepared by serial dilutions in solvent control solution.

Images were analyzed using ImageJ software. For AWA imaging, the fluorescence change in the soma was measured and selected as the region of interest. The fluorescence of an equally sized region of interest was captured from the background. The fluorescence of the background was then subtracted from the neuron for each frame to obtain background-subtracted fluorescence. $F/F_0$ was then calculated by dividing each frame by $F_0$ (average fluorescence from seconds 2–3 of each trial). This adjusted $F/F_0$ value was corrected to be the percent change in fluorescence by the following equation: $(F/F_0 - 1) * 100\%$.

The maximum change in fluorescence was calculated for each trial. The maximum value for the 'Pre' is defined as the maximum percent change in fluorescence from 0.0 to 4.9 s. The 'Stim' period was 5.0–15.0 s. The 'Post' period was 15.1–29.9 s.

## Acknowledgements

We thank Donovan Ventimiglia, Paul Greer, Matthew Lovett-Barron, Eviatar Yemini, Taralyn Tan, and members of the Flavell lab for critical reading of the manuscript. We thank Mei Zhen, Michael O'Donnell, and Piali Sengupta for plasmids, and the Caenorhabditis Genetics Center (supported by P40 OD010440), Horvitz lab, and Bargmann lab for strains. We thank the MIT BioMicroCenter for RNA library preparation and sequencing. IGM was supported by the Picower Fellows program. JS acknowledges funding from NIH (DC016058). SWF acknowledges funding from the JPB Foundation, NIH (NS104892), NSF (#1845663), McKnight Foundation, and Alfred P Sloan Foundation.

## Additional information

### Funding

| Funder | Grant reference number | Author |
| --- | --- | --- |
| National Institute on Deafness and Other Communication Disorders | DC016058 | Jagan Srinivasan |
| JPB Foundation | | Steven W Flavell |
| National Institute of Neurological Disorders and Stroke | NS104892 | Steven W Flavell |
| National Science Foundation | 1845663 | Steven W Flavell |
| McKnight Endowment Fund for Neuroscience | | Steven W Flavell |
| Alfred P. Sloan Foundation | | Steven W Flavell |

The funders had no role in study design, data collection and interpretation, or the decision to submit the work for publication.

## Author contributions
Ian G McLachlan, Talya S Kramer, Conceptualization, Formal analysis, Investigation, Methodology, Writing – original draft, Writing – review and editing; Malvika Dua, Formal analysis, Investigation, Writing – original draft, Writing – review and editing; Elizabeth M DiLoreto, Investigation, Methodology, Writing – review and editing; Matthew A Gomes, Investigation; Ugur Dag, Investigation, Writing – review and editing; Jagan Srinivasan, Supervision, Funding acquisition, Methodology, Writing – review and editing; Steven W Flavell, Conceptualization, Supervision, Funding acquisition, Investigation, Methodology, Writing – original draft, Writing – review and editing

## Author ORCIDs
Ian G McLachlan ⓘ http://orcid.org/0000-0002-1897-7288
Talya S Kramer ⓘ http://orcid.org/0000-0002-3763-511X
Ugur Dag ⓘ http://orcid.org/0000-0001-6937-5722
Jagan Srinivasan ⓘ http://orcid.org/0000-0001-5449-7938
Steven W Flavell ⓘ http://orcid.org/0000-0001-9464-1877

## Decision letter and Author response
Decision letter https://doi.org/10.7554/eLife.79557.sa1
Author response https://doi.org/10.7554/eLife.79557.sa2

## Additional files

### Supplementary files
• Supplementary file 1. List of neuronal genes whose expression is altered by fasting. Table of genes that are upregulated or downregulated in neurons in fasted animals. Transcripts from neuronal ribotagging biological replicates (three per condition, fed versus fasted) were mapped to the *C. elegans* genome (WBcel235) with kallisto then analyzed for differential expression of annotated genes with sleuth. Sleuth generated a gene-level model fit to ribotagged vs. input and fed vs. fasted conditions. A Wald test was then applied to generate a beta statistic (b), which approximates to the log2 fold-change in expression between the fed and starved conditions. Genes were included if b > 2 (upregulated in fasted, 802 genes) or b < -2 (downregulated in fasted, 647 genes).

• MDAR checklist

### Data availability
All data are publicly available. RNA sequencing data is deposited at GEO accession number GSE200640. All other data (fluorescent reporter and calcium imaging data) is deposited on Dryad at https://doi.org/10.5061/dryad.t4b8gtj4h.

The following datasets were generated:

| Author(s) | Year | Dataset title | Dataset URL | Database and Identifier |
|---|---|---|---|---|
| McLachlan I, Dua M, Flavell S | 2022 | Diverse states and stimuli tune olfactory receptor expression levels to modulate food-seeking behavior | https://dx.doi.org/10.5061/dryad.t4b8gtj4h | Dryad Digital Repository, 10.5061/dryad.t4b8gtj4h |
| McLachlan I, Flavell SW | 2022 | Pan-neuronal ribotagging in fasted *C. elegans* | https://www.ncbi.nlm.nih.gov/geo/query/acc.cgi?acc=GSE200640 | NCBI Gene Expression Omnibus, GSE200640 |

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
