## [Editor Report]

This article provides a detailed explanation of how *C. elegans* adapts its behavior and chemosensory responses to shifts in food availability. The authors show that prolonged fasting broadly alters gene expression in food-sensing neurons, thereby altering foraging behavior and chemosensory responses to food. The fasting-induced genes include many chemoreceptors, one of which mediates responses to specific volatile components of food. Finally, they show that food controls expression of a fasting-induced chemoreceptor via multiple external (i.e., sensory) and internal (potentially metabolic) cues.

---

## [Decision Letter]

**Decision letter after peer review:**

Thank you for submitting your article "Diverse states and stimuli tune olfactory receptor expression levels to modulate food-seeking behavior" for consideration by *eLife*. Your article has been reviewed by 3 peer reviewers, and the evaluation has been overseen by a Reviewing Editor and Ronald Calabrese as the Senior Editor. The following individual involved in the review of your submission has agreed to reveal their identity: Mike Crickmore (Reviewer #1).

Essential revisions:

The reviewers consider the work well executed and nicely presented. Major concerns address a few outstanding issues to clarify or strengthen specific points in the manuscript, all of which should be easily achieved.

1. The authors provide several results supporting the idea that increased str-44 and srd-28 expression drive fasting-induced changes in AWA chemosensory responses to food and foraging behavior. The authors should revise the text to indicate that other fasting regulated AWA genes may also contribute to these changes. Alternatively, if they want to more strongly implicate STR-44 and SRD-28's function, they should analyze the effect of mutations inactivating these chemoreceptors on fasting-induced changes in AWA responses and foraging behavior.

2. The authors argue that the presence of food inhibits STR-44 expression via increased insulin signaling. Specifically, they suggest that a specific insulin-like ligand (DAF-28) mediates the inhibitory effects of food on STR-44 expression (Figure 4C). These results should be interpreted more cautiously. The daf-28(sa191) mutation used in this analysis is not a loss of function allele. Instead, this dominant allele constitutively induces the unfolded protein response in the ASI neuron (Kulalurt, 2013). Thus, the phenotype observed in this strain may not result from a simple loss of ASI secretion of DAF-28. This analysis also does not indicate which cell type (ASI or other neurons) produces DAF-28 to inhibit STR-44 expression.

3. DAF-16/FOXO transcriptional activity in AWA is argued to be a node of convergence for multiple signals regulating STR-44 expression. However, transgenes expressing DAF-16 selectively in AWA did not fully rescue the daf-16 mutant defects (Figure 4C). The authors should revise the text to indicate that DAF-16 may function in both AWA and other cell types to regulate str-44 expression. Alternatively, the author could strengthen the DAF-16 site of action analysis by using other techniques for AWA-specific knockout, knockdown, or proteolytic degradation.

4. The authors distinguish between different components of food (volatile odors, taste, mechanosensory responses, and nutrient value) by using different food sources, inedible (aztreonam treated) food, and exposure to food on the lid of petri dishes. While generally supportive of their conclusions, the authors should describe other potential interpretations of these results. For example, the sensory experience of chemically treated bacteria or food on the petri lid may not be strictly identical to swimming in a bacterial lawn.

5. Throughout the text, the authors propose that food inhibition of STR-44 expression is a consequence of convergent food-associated chemosensory cues and internal state (i.e. hunger or metabolic changes). It would be helpful for the readers if the authors more clearly explained these competing ideas for how food regulates STR-44 expression and behavior. Which specific phenotypes suggest that worms exhibit "hunger" rather than long-term sensory adaptation to food deprivation? Which results most strongly support the idea of an internal state regulating STR-44 expression (rictor function in the intestine)? Which results most strongly indicate that fasting-induced metabolic changes are relevant?

*Reviewer #1 (Recommendations for the authors):*

I don't love rescaling the Y-axis, but see why the authors feel the need to do it. On some graphs (ex: 4C, 6B), the large y-axis takes away from smaller changes in str-44 expression. It may be worth splitting up the graphs to make it easier to appreciate these smaller changes.

*Reviewer #2 (Recommendations for the authors):*

First, the authors should be congratulated for this carefully conducted, ambitious study engaging an impressive set of methodologies and a well-written article that was very interesting to read. Here below are a few specific points that should be addressed to clarify the manuscript and potentially further strengthen the scientific conclusions:

1. Since a major claim of the proposed signalling model is to place DAF-16 as a downstream convergence node of multiple pathways, the authors should determine if DAF-16 works only in AWA for regulating str-44 expression. At present, this view is supported by an AWA-specific rescue of daf-16 mutation. However, the effect seems to be partial, which leaves open the possibility that DAF-16 might also work in additional tissues. The authors should consider the following two experiments: (i) use a control rescue with daf-16 promoter, to see if a full rescue can be obtained; (ii) perform an AWA-specific knockdown of daf-16.

2. Data in Figure 3 Supplement 1 and Figure 4B, should be presented and statistically analysed together, in order to highlight the fasting effect. The opposing behavior of ASI in fed versus food deprived will then need to be discussed (and the scheme in Figure 4G, reconsidered).

3. Line 40, remove 'recent' as long-lasting internal states could reflect experiences across timescales.

4. Figure 1F, FigS1F: Without the experiment with inedible bacteria for various strains the authors cannot dissect the contribution of odor and metabolic components. In line 195, 'metabolic' word should be removed because the effect on food could be due to nutrient value plus touch (sensory effect) of food. In addition, the authors should keep in mind (and warn the reader) that it is virtually impossible to judge if the sensory experience of an odor coming from the lid or from the contacting bacterial mass is really identical. Even if no convincing alternative is available, it is perhaps still slightly regrettable that aztreonam-based and related experiments aiming at dissecting the mode of action of food are becoming standards in the field, because most of them are very difficult to control and interpret. The conclusions need to be phrased very cautiously.

5. To conclusively show that STR-44 and SRD-28 detect odors in AWA, authors should perform calcium imaging of AWA in response to food or relevant odors either in freely moving or immobilized worms, comparing wt and these GPCR mutants.

6. Figure 4C, 5B, 6A, 6C: the representation of statistical comparison is confusing and should be improved.

7. Figure 4E: the authors partially blocked str-44 induction when inhibiting AWA activity using HisCl channel. This indicates that AWA depolarisation is partially required, but cannot distinguish between direct effects of autonomous cell activity (including, e.g., ensuing intracellular calcium signaling) and indirect effects implicating cell-to-cell communication and a potential feedback loop (such as via AIA). To discriminate between these two possibilities, the authors should also block synaptic transmission of AWA using TeTx to dissect potential feedback mechanisms that could change str-44 expression.

8. Figure 4G: no data are provided that support the notion that DAF-28 actually originates from ASI in the regulation of str-44. Figure and text should be revised accordingly, or AWA-specific daf-28 manipulation experiments should be carried out.

9. Line 388, authors should refer to the data source that was used to conclude about AWA connectivity.

10. Most of the mechanisms dissected in the study act to inhibit the increase of str-44 expression on food. The result of raga-1 mutation is particularly intriguing. Do authors think TORC-1 dependent positive signal could alter the expression of str-44 after starvation? To further dissect the contribution of TORC-1 vs TORC-2 authors can perform a choice test experiment (Figure 5D in rict-1, raga-1 fed and starved condition). Authors should at least discuss the contribution of "positively acting" pathways that may be activated upon starvation. At present, the manuscript seems to be somewhat biased toward inhibitory pathways engaged in food.

11. The 'lateral signaling' nomenclature in this context, even if broadly and explicitly defined by the authors, might be misleading. Indeed, the impact of sensory neurons could be largely mediated by ascending feedback from AIA (it would be unconventional to consider AIA to be lateral to AWA). There are precedents for this type of ascending feedback (see PMC2937567, for example), and no data in the present study seems to exclude this possibility. The fact that AIA::TeTx nearly abolishes fasting-evoked str-44 expression increase (perhaps the most effective cell-specific manipulation reported in the article) is also compatible with this view. Actually, it seems that the layout of the model in Figure 4G was created with this idea in mind. The authors should provide a deeper discussion about this aspect, and re-consider the use of the 'lateral signaling' terminology, which evokes a more direct mode of action. In this context, the experiment suggested above under point 7 might be really helpful.

*Reviewer #3 (Recommendations for the authors):*

The authors conclude that str-44 is a chemoreceptor for propyl acetate and butyl acetate, the conclusion could be strengthened by conducting the chemosensory assays using an str-44 mutant which would show that mutations only impair propyl acetate and butyl acetate attraction.

Similarly, using str-44 mutants for the calcium imaging experiments would strengthen the conclusion that the responses to cues detected by the receptor is responsible for the increase in activity in AWA of fasted animals (Figure 2C, D).

---

## [Author Response]

Essential revisions:The reviewers consider the work well executed and nicely presented. Major concerns address a few outstanding issues to clarify or strengthen specific points in the manuscript, all of which should be easily achieved.1. The authors provide several results supporting the idea that increased str-44 and srd-28 expression drive fasting-induced changes in AWA chemosensory responses to food and foraging behavior. The authors should revise the text to indicate that other fasting regulated AWA genes may also contribute to these changes. Alternatively, if they want to more strongly implicate STR-44 and SRD-28's function, they should analyze the effect of mutations inactivating these chemoreceptors on fasting-induced changes in AWA responses and foraging behavior.

We substantially revised the text to indicate that other upregulated genes may contribute to fasting induced changes in AWA activity and animal behavior, wherever relevant.

2. The authors argue that the presence of food inhibits STR-44 expression via increased insulin signaling. Specifically, they suggest that a specific insulin-like ligand (DAF-28) mediates the inhibitory effects of food on STR-44 expression (Figure 4C). These results should be interpreted more cautiously. The daf-28(sa191) mutation used in this analysis is not a loss of function allele. Instead, this dominant allele constitutively induces the unfolded protein response in the ASI neuron (Kulalurt, 2013). Thus, the phenotype observed in this strain may not result from a simple loss of ASI secretion of DAF-28. This analysis also does not indicate which cell type (ASI or other neurons) produces DAF-28 to inhibit STR-44 expression.

We thank the reviewers for identifying this important caveat. We performed additional experiments with a *daf-28(tm2308)* deletion allele and found that it did not phenocopy *sa191*. We have decided to remove the DAF-28 result from the manuscript.

3. DAF-16/FOXO transcriptional activity in AWA is argued to be a node of convergence for multiple signals regulating STR-44 expression. However, transgenes expressing DAF-16 selectively in AWA did not fully rescue the daf-16 mutant defects (Figure 4C). The authors should revise the text to indicate that DAF-16 may function in both AWA and other cell types to regulate str-44 expression. Alternatively, the author could strengthen the DAF-16 site of action analysis by using other techniques for AWA-specific knockout, knockdown, or proteolytic degradation.

We performed a new experiment in which we performed pan-body (P*dpy-30*) rescue of DAF-16A, the isoform that was also used for AWA-specific rescue. We found that this transgene also does not fully rescue the *daf-16* mutant phenotype. We have revised the manuscript to indicate that multiple isoforms of DAF-16 are likely required and that we have not ruled out the possibility that DAF-16 may function in other cell types in addition to AWA. [Figure 4C, lines 398-403]

4. The authors distinguish between different components of food (volatile odors, taste, mechanosensory responses, and nutrient value) by using different food sources, inedible (aztreonam treated) food, and exposure to food on the lid of petri dishes. While generally supportive of their conclusions, the authors should describe other potential interpretations of these results. For example, the sensory experience of chemically treated bacteria or food on the petri lid may not be strictly identical to swimming in a bacterial lawn.

We provided the caveats inherent in using aztreonam to produce inedible bacteria and added a more cautious interpretation of the results. [lines 174-191]

5. Throughout the text, the authors propose that food inhibition of STR-44 expression is a consequence of convergent food-associated chemosensory cues and internal state (i.e. hunger or metabolic changes). It would be helpful for the readers if the authors more clearly explained these competing ideas for how food regulates STR-44 expression and behavior. Which specific phenotypes suggest that worms exhibit "hunger" rather than long-term sensory adaptation to food deprivation? Which results most strongly support the idea of an internal state regulating STR-44 expression (rictor function in the intestine)? Which results most strongly indicate that fasting-induced metabolic changes are relevant?

We revised the text throughout the manuscript to emphasize whether the effects of internal nutritional state versus chemosensory experience were being tested. We also directly addressed a potential sensory habituation hypothesis proposed by Reviewer #1 by examining whether exposure to butyl or propyl acetate (the STR-44 ligands) alter *str-44* expression. We found that they do not. [Figure 4—figure supplement 2B; lines 414-421] Thus, we also revised a paragraph in the discussion indicating why we view an integrative hypothesis as most parsimonious with our data. [lines 557-568] In addition, we modified the abstract and introduction to balance our emphasis on internal state and sensory experience more equally.

Reviewer #1 (Recommendations for the authors):I don't love rescaling the Y-axis, but see why the authors feel the need to do it. On some graphs (ex: 4C, 6B), the large y-axis takes away from smaller changes in str-44 expression. It may be worth splitting up the graphs to make it easier to appreciate these smaller changes.

Unfortunately, because of the magnitude of the changes in some mutants (namely *tax-4* and *daf-2*), we have no choice but to re-scale to accommodate them; carrying these large scales forward to other panels would compound the problem that we see in panels 4C and 6A-C. We explored whether we could split up conditions with higher expression, but we thought that it is preferable to keep them together to facilitate comparison between these conditions, with the unfortunate effect that it makes certain significant changes appear small.

Reviewer #2 (Recommendations for the authors):First, the authors should be congratulated for this carefully conducted, ambitious study engaging an impressive set of methodologies and a well-written article that was very interesting to read. Here below are a few specific points that should be addressed to clarify the manuscript and potentially further strengthen the scientific conclusions:1. Since a major claim of the proposed signalling model is to place DAF-16 as a downstream convergence node of multiple pathways, the authors should determine if DAF-16 works only in AWA for regulating str-44 expression. At present, this view is supported by an AWA-specific rescue of daf-16 mutation. However, the effect seems to be partial, which leaves open the possibility that DAF-16 might also work in additional tissues. The authors should consider the following two experiments: (i) use a control rescue with daf-16 promoter, to see if a full rescue can be obtained; (ii) perform an AWA-specific knockdown of daf-16.

We addressed this concern by performing a control rescue using the broadly expressed promoter *dpy-30* to drive DAF-16A expression. We found that the rescue effect is still partial relative to wild-type controls. We anticipate that the partial rescue is due to the use of only the DAF-16A isoform. We have amended the text to indicate that multiple *daf-16* isoforms are likely required and that we have not excluded the possibility that *daf-16* acts in multiple tissues. [Figure 4C, lines 398-403]

2. Data in Figure 3 Supplement 1 and Figure 4B, should be presented and statistically analysed together, in order to highlight the fasting effect. The opposing behavior of ASI in fed versus food deprived will then need to be discussed (and the scheme in Figure 4G, reconsidered).

We believe that the fasted data for the HisCl manipulations is confusing to include in the main figure and detracts from the primary point that a broad set of sensory neurons act in fed animals to suppress *str-44* expression in the fed state. In addition, due to the variability in the fasted controls without histamine (Figure 4—figure supplement 1), we are hesitant to overinterpret this data. However, we agree with the reviewer that the ASI result is worth highlighting, and so we have modified the text to do so. [lines 376378]

3. Line 40, remove 'recent' as long-lasting internal states could reflect experiences across timescales.

We made this correction.

4. Figure 1F, FigS1F: Without the experiment with inedible bacteria for various strains the authors cannot dissect the contribution of odor and metabolic components. In line 195, 'metabolic' word should be removed because the effect on food could be due to nutrient value plus touch (sensory effect) of food. In addition, the authors should keep in mind (and warn the reader) that it is virtually impossible to judge if the sensory experience of an odor coming from the lid or from the contacting bacterial mass is really identical. Even if no convincing alternative is available, it is perhaps still slightly regrettable that aztreonam-based and related experiments aiming at dissecting the mode of action of food are becoming standards in the field, because most of them are very difficult to control and interpret. The conclusions need to be phrased very cautiously.

We agree that aztreonam treatment is an imperfect experiment and have added more cautious language to this section. [lines 174-191]

5. To conclusively show that STR-44 and SRD-28 detect odors in AWA, authors should perform calcium imaging of AWA in response to food or relevant odors either in freely moving or immobilized worms, comparing wt and these GPCR mutants.

We agree that this would be required to conclusively show that STR-44 and SRD-28 are necessary for the increased calcium response to food or odor in fasted animals. However, we have avoided making the claims that either receptor is strictly necessary for this effect, as we expect some degree of redundancy among GPCRs. This is because while the mutant does have a behavioral phenotype, it is mild relative to the overexpression phenotype, and because fed animals detect and are attracted to butyl acetate. We have clarified in the text that other changes in AWA may be involved. See also: response to Essential Revision #1.

6. Figure 4C, 5B, 6A, 6C: the representation of statistical comparison is confusing and should be improved.

We made this correction.

7. Figure 4E: the authors partially blocked str-44 induction when inhibiting AWA activity using HisCl channel. This indicates that AWA depolarisation is partially required, but cannot distinguish between direct effects of autonomous cell activity (including, e.g., ensuing intracellular calcium signaling) and indirect effects implicating cell-to-cell communication and a potential feedback loop (such as via AIA). To discriminate between these two possibilities, the authors should also block synaptic transmission of AWA using TeTx to dissect potential feedback mechanisms that could change str-44 expression.

Thank you for this suggestion. We carried out this experiment and included it in Figure 4—figure supplement 2. AWA::TeTx has no effect on fed or starved expression of *str-44*, suggesting that the former hypothesis (direct effects of autonomous cell activity) is more likely. [Figure 4—figure supplement 2; lines 433-439]

8. Figure 4G: no data are provided that support the notion that DAF-28 actually originates from ASI in the regulation of str-44. Figure and text should be revised accordingly, or AWA-specific daf-28 manipulation experiments should be carried out.

We performed experiments with an additional *daf-28* allele that changed our interpretations in this section. A full response to this point is above, in response to Essential Revision #2.

9. Line 388, authors should refer to the data source that was used to conclude about AWA connectivity.

We added this information and clarified the connectivity.

10. Most of the mechanisms dissected in the study act to inhibit the increase of str-44 expression on food. The result of raga-1 mutation is particularly intriguing. Do authors think TORC-1 dependent positive signal could alter the expression of str-44 after starvation? To further dissect the contribution of TORC-1 vs TORC-2 authors can perform a choice test experiment (Figure 5D in rict-1, raga-1 fed and starved condition). Authors should at least discuss the contribution of "positively acting" pathways that may be activated upon starvation. At present, the manuscript seems to be somewhat biased toward inhibitory pathways engaged in food.

We agree that the *raga-1* phenotype is interesting and worth further investigation. We felt that extending a single manuscript beyond the inhibitory pathways engaged on food would become unwieldy, but we are excited to continue work on pathways activated off food. We highlighted the *raga-1* result with a note about the possibility of additional “positively acting” pathways. [lines 463-465]

11. The 'lateral signaling' nomenclature in this context, even if broadly and explicitly defined by the authors, might be misleading. Indeed, the impact of sensory neurons could be largely mediated by ascending feedback from AIA (it would be unconventional to consider AIA to be lateral to AWA). There are precedents for this type of ascending feedback (see PMC2937567, for example), and no data in the present study seems to exclude this possibility. The fact that AIA::TeTx nearly abolishes fasting-evoked str-44 expression increase (perhaps the most effective cell-specific manipulation reported in the article) is also compatible with this view. Actually, it seems that the layout of the model in Figure 4G was created with this idea in mind. The authors should provide a deeper discussion about this aspect, and re-consider the use of the 'lateral signaling' terminology, which evokes a more direct mode of action. In this context, the experiment suggested above under point 7 might be really helpful.

As suggested, we removed all references to “lateral signaling” and replaced them with “sensory crosstalk” or simply “signaling.” In addition, we cited previous studies that also found important roles for ascending feedback from AIA.

Reviewer #3 (Recommendations for the authors):The authors conclude that str-44 is a chemoreceptor for propyl acetate and butyl acetate, the conclusion could be strengthened by conducting the chemosensory assays using an str-44 mutant which would show that mutations only impair propyl acetate and butyl acetate attraction.

This result was included in the submitted manuscript as Figure S2D, which shows that *str-44;srd-28* animals do not exhibit a significant increase in butyl acetate preference when fasted. We placed this in the supplement because it is a mild effect and we did not wish to claim that STR-44 and SRD-28 are the only chemoreceptors that may be involved in this behavior. See also: response to Essential Revision #1 and comment below.

Similarly, using str-44 mutants for the calcium imaging experiments would strengthen the conclusion that the responses to cues detected by the receptor is responsible for the increase in activity in AWA of fasted animals (Figure 2C, D).

We agree that this would be required to conclusively show that STR-44 and SRD-28 are necessary for the increased calcium response to food or odor in fasted animals. However, we have avoided making the claims that either receptor is strictly necessary for this effect, as we expect some degree of redundancy among GPCRs. This is because while the mutant does have a behavioral phenotype, it is mild relative to the overexpression phenotype, and because fed animals detect and are attracted to butyl acetate. We have clarified in the text that other changes in AWA may be involved. See also: response to Essential Revision #1 and comment above.